# Joint Metric Space Embedding by Unbalanced Optimal Transport with Gromov—Wasserstein Marginal Penalization

**Florian Beier** [* 1] **Moritz Piening** [* 1] **Robert Beinert** [1] **Gabriele Steidl** [1]

## Abstract

We propose a new approach for unsupervised alignment of heterogeneous datasets, which maps data from two different domains without any known correspondences to a common metric space. Our method is based on an unbalanced optimal transport problem with Gromov–Wasserstein marginal penalization. It can be seen as a counterpart to the recently introduced joint multidimensional scaling method. We prove that there exists a minimizer of our functional and that for penalization parameters going to infinity, the corresponding sequence of minimizers converges to a minimizer of the so-called embedded Wasserstein distance. Our model can be reformulated as a quadratic, multi-marginal, unbalanced optimal transport problem, for which a bi-convex relaxation admits a numerical solver via block-coordinate descent. We provide numerical examples for joint embeddings in Euclidean as well as non-Euclidean spaces.

## 1. Introduction

The comparison of heterogeneous data distributions is a fundamental task in computer vision, computational biology and machine learning. Most existing approaches rely on using a suitable ground cost function such as an available metric. A classic example is the Wasserstein distance which seeks an optimal transport (OT) between two given distributions. However, often the given distributions are in heterogeneous spaces, where a readily available ground cost function between these spaces does not generally exist. Additional effort may be required to learn appropriate cost functions (Cuturi & Avis, 2014; Heitz et al., 2021). However, even if there exists a natural embedding into a

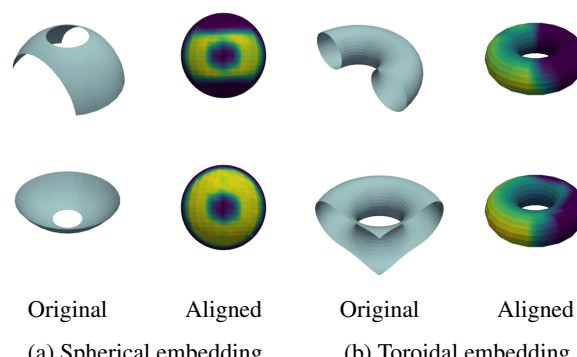

Original    Aligned    Original    Aligned

(a) Spherical embedding    (b) Toroidal embedding

*Figure 1.* Joint (aligning) transfer of two metric spaces (gray surfaces with surface distance) to a fixed reference space, namely to the sphere and the torus by our method, where the color "yellow" corresponds to higher values, see Subsection 6.1.

canonical joint metric space, this metric may not be suited to accurately gauge differences in their samples. Examples of such heterogeneous settings are, e.g., the comparison of graph- or mesh-valued data such as 3d shapes or manifolds. This paper introduces a novel framework based on OT which enables the joint comparison and visualization of heterogeneous datasets by optimally transferring them into an a-priori fixed metric space, see Figure 1 for an illustrative example.

### 1.1. Previous Work

As a first step, we highlight the most relevant contributions related to our study among the vast literature on OT and dimensionality reduction.

**Optimal Transport with Invariances** Classical Wasserstein distances allow to compare measures on a common metric space. In specific tasks in geometry processing like shape matching, the considered measures, however, live on distinct metric spaces that incorporate the different geodesic distances on the given shapes. To address this, Gromov–Wasserstein (GW) distances are introduced (Mémoli, 2011; Sturm, 2006). In (Alaya et al., 2022), an approximation of GW distances is obtained by jointly embedding measures into Euclidean spaces. Furthermore, projection- and

---

*Equal contribution [1]Institut für Mathematik, Technische Universität Berlin, Germany. Correspondence to: Moritz Piening <piening@math.tu-berlin.de>.

*Proceedings of the 42nd International Conference on Machine Learning*, Vancouver, Canada. PMLR 267, 2025. Copyright 2025 by the author(s).

subspace-robust Wasserstein-2 distances are introduced in (Paty & Cuturi, 2019). Another approach incorporates invariance to Euclidean isometries via the Wasserstein Procrustes problem (Grave et al., 2019). This is extended to account for linear operators with bounded Schatten norms in (Alvarez-Melis et al., 2019) and to Gaussian mixture applications in (Salmona et al., 2024). As outlined in Section 5, our paper extends such invariant OT to non-Euclidean domains.

**Joint Dimensionality Reduction** Dimensionality reduction is a core topic in machine learning enabling visualization and clustering by finding optimal low-dimensional representations of data. Classical methods like principal component analysis (PCA) (Greenacre et al., 2022) and multidimensional scaling (MDS) (Carroll & Arabie, 1998) preserve large variations or pairwise distances, but fail on nonlinear manifolds (Alaya et al., 2022; Deng et al., 2024). Nonlinear approaches, e.g., locally linear embedding (Roweis & Saul, 2000), probabilistic models, e.g., t-distributed stochastic neighbor embedding (t-SNE) (Van der Maaten & Hinton, 2008), and deep learning methods, e.g., variational autoencoders (VAEs), (Kingma et al., 2019) address these challenges. As an extension, several recent methods focus on joint embeddings of heterogeneous data. Here, we are given data on two incompatible domains and are interested in simultaneous embedding. The manifold-aligning generative adversarial network (Amodio & Krishnaswamy, 2018) employs a generative adversarial network for domain alignment. Maximum mean discrepancy (MMD) manifold-alignment (Liu et al., 2019) balances an MMD and a distortion term. UnionCom (Cao et al., 2020) leverages the generalized unsupervised manifold alignment (GUMA) (Cui et al., 2014). Single-cell alignment with optimal transport (SCOT) (Demetci et al., 2022) and the partial manifold alignment algorithm (Cao et al., 2022) employ GW distances. Finally, the recently proposed joint multidimensional scaling (JMDS) (Chen et al., 2023) algorithm combines MDS with OT for the joint embedding of two datasets into a shared Euclidean space. Complementing this and research on non-Euclidean embeddings (McInnes et al., 2018; Deng et al., 2024) and connections between MDS and GW (Van Assel et al., 2024; Clark et al., 2025), we propose a joint embedding into arbitrary metric spaces based on factored transport plans (Forrow et al., 2019) and GW distances.

### 1.2. Contribution

Our main contributions are the following:
1. Towards the aligned "embedding" of heterogeneous metric spaces into a fixed, not necessarily Euclidean space, we propose to minimize an unbalanced OT problem with a quadratic cost function, where the marginals are penalized by GW distances. This formulation seeks near-isometric joint embeddings of the inputs into a metric space, while enabling an optimal comparison in the Wasserstein distance.
2. We prove that our functional has a minimizer. Further, if its regularization parameter goes to infinity, our functional approaches the so-called "embedded Wasserstein distance" (Salmona et al., 2024). In this sense, we also refer to our model as the "relaxed embedded Wasserstein distance".
3. For the (approximate) computation of a minimizer, we provide an equivalent formulation of our model as a quadratic, multi-marginal, unbalanced OT problem. A biconvex relaxation enables the application of existing algorithms to solve the problem numerically.
4. We recall JMDS from the point of view of our model: while we are searching for the weight of atomic measures fixing their supports, JMDS fixes the weights and aims to find the supports of the measures.
5. Numerical experiments demonstrate the potential of our method for joint embeddings of heterogeneous data on the 2d Euclidean space, the torus, the 2-sphere and the space of 2d Gaussians with the Wasserstein distance.

All proofs are given in Appendix A.

## 2. OT-Based Distances

Given a compact metric space $(Z, d_Z)$, we denote by $\mathcal{P}(Z)$ the space of probability measures defined on the Borel-$\sigma$-algebra induced by the distance $d_Z$. For two compact metric spaces $Z_1$ and $Z_2$, the *push-forward measure* of $\mu \in \mathcal{P}(Z_1)$ under a measurable map $T \colon Z_1 \to Z_2$ is denoted by

$$T_\sharp \mu \coloneqq \mu \circ T^{-1} \in \mathcal{P}(Z_2).$$

The *set of transport plans* between two measures $\mu_1 \in \mathcal{P}(Z_1)$ and $\mu_2 \in \mathcal{P}(Z_2)$ is given by

$$\Pi(\mu_1, \mu_2) \coloneqq \big\{ \pi \in \mathcal{P}(Z_1 \times Z_2) : P_{i,\sharp}\pi = \mu_i, \; i = 1, 2 \big\},$$

where $P_i \colon Z_1 \times Z_2 \to Z_i, (z_1, z_2) \mapsto z_i, i = 1, 2$, denotes the projection to the first and second component, respectively.

To lift $d_Z$ from $Z$ to $\mathcal{P}(Z)$, we rely on the *Wasserstein(-2) distance* between $\mu_1, \mu_2 \in \mathcal{P}(Z)$ defined by

$$\mathrm{W}(\mu_1, \mu_2) \coloneqq \min_{\pi \in \Pi(\mu_1, \mu_2)} \Big( \int_{Z \times Z} d_Z^2(z, z') \, \mathrm{d}\pi(z, z') \Big)^{\frac{1}{2}}. \quad (1)$$

By $\Pi_\mathrm{o}(\mu_1, \mu_2)$, we denote the set of minimizers in (1). The Wasserstein distance metricizes the weak convergence of measures, where a sequence of measures $(\mu_n)_{n \in \mathbb{N}}$ converges weakly to a measure $\mu \in \mathcal{P}(Z)$, written $\mu_n \rightharpoonup \mu$, if for every (bounded) continuous function $\varphi \colon Z \to \mathbb{R}$, we have $\int_Z \varphi \, \mathrm{d}\mu_n \to \int_Z \varphi \, \mathrm{d}\mu$ as $n \to \infty$.

To compare heterogeneous data via their internal geometry, we can rely on GW distances, which enables us to compare

measures on different metric spaces. To highlight the connection between measures and underlying spaces, we consider *metric measure spaces (mm-spaces)*, which are triples $\mathbb{X} = (X, d_X, \xi)$ such that $(X, d_X)$ is a compact metric space and $\xi \in \mathcal{P}(X)$. Two mm-spaces $\mathbb{X}_i = (X_i, d_{X_i}, \xi_i)$, $i = 1, 2$, are *isomorphic* if there exists a bijective map $I \colon \operatorname{supp} \xi_1 \to \operatorname{supp} \xi_2$ between the supports of the measures such that $I$ is *measure-preserving*, i.e., $I_\sharp \xi_1 = \xi_2$, and $I$ is an *isometry*, i.e., $d_{X_1}(x_1, x_1') = d_{X_2}(I(x_1), I(x_1'))$ for all $x_1, x_1' \in \operatorname{supp} \xi_1$. By $[\mathbb{X}]$ we denote the equivalence class of all mm-spaces which are isomorphic to $\mathbb{X}$. Finally, if $I$ is an isometry between two metric spaces $X_i$, $i = 1, 2$, we write $I \colon X_1 \hookrightarrow X_2$, and just $X_1 \hookrightarrow X_2$ if such an isometry between $X_1$ and $X_2$ exists.

*Sturm's GW distance* (Sturm, 2006) between two mm-spaces $\mathbb{X}_i$, $i = 1, 2$, is defined by

$$\operatorname{SGW}(\mathbb{X}_1, \mathbb{X}_2) := \inf_{\substack{(Z, d_Z) \text{ metric space,} \\ I_i \colon \operatorname{supp} \xi_i \hookrightarrow Z, i = 1, 2}} \operatorname{W}(I_{1,\sharp}\xi_1, I_{2,\sharp}\xi_2),$$

where the Wasserstein distance is taken over the respective space $(Z, d_Z)$. The SGW distance seeks optimal isometric embeddings of either mm-space into a joint metric space such that their Wasserstein distance in the embedded space is minimal, see Figure 2a. Indeed, SGW is a metric on the equivalence classes of mm-spaces.

Mémoli proposed a different GW distance, which is numerically more appealing than Sturm's construction. For two mm-spaces $\mathbb{X}_i$, $i = 1, 2$, *Mémoli's GW distance* (Mémoli, 2011) is defined by

$$\operatorname{GW}(\mathbb{X}_1, \mathbb{X}_2) := \min_{\gamma \in \Pi(\xi_1, \xi_2)} G_{X_1, X_2}(\gamma)^{\frac{1}{2}}$$

with the quadratic GW objective

$$G_{X_1, X_2}(\gamma) := \iint_{(X_1 \times X_2)^2} \left( d_{X_1}(x_1, x_1') - d_{X_2}(x_2, x_2') \right)^2 \\ \times \mathrm{d}\gamma(x_1, x_2)\,\mathrm{d}\gamma(x_1', x_2').$$

The problem seeks a transport between $\mathbb{X}_1$ and $\mathbb{X}_2$ which minimizes the overall pairwise distance distortion, see Figure 2b. Both distances—SGW and GW— define a metric on the equivalence classes of mm-spaces. Furthermore, both distances induce the same topology, but SGW turns these equivalence classes into a complete metric space, while GW does not (Sturm, 2006; Mémoli, 2011). The later can be alleviated by embedding into a larger space and extending the GW definition accordingly (Sturm, 2023).

## 3. Unbalanced OT With GW Penalization

Throughout this section, let $(Z, d_Z)$ be a fixed compact metric space. Furthermore, let $\mathbb{X}_i = (X_i, d_{X_i}, \xi_i)$, $i =$

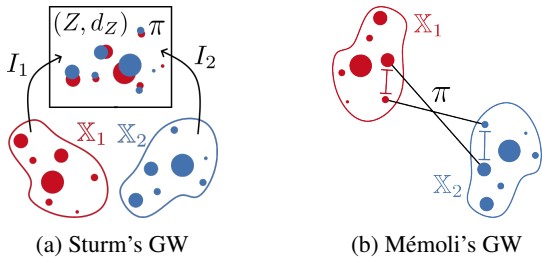

(a) Sturm's GW          (b) Mémoli's GW

*Figure 2.* Illustration of GW formulations of Sturm and Mémoli.

$1, 2$, be two mm-spaces, whose measures' support can be isometrically embedded into $(Z, d_Z)$, i.e., $\operatorname{supp} \xi_i \hookrightarrow Z$. For this specific setting, we relax Sturm's GW distance to

$$\operatorname{EW}(\mathbb{X}_1, \mathbb{X}_2) := \inf_{\substack{I_1 \colon \operatorname{supp} \xi_1 \hookrightarrow Z \\ I_2 \colon \operatorname{supp} \xi_2 \hookrightarrow Z}} \operatorname{W}(I_{1,\sharp}\xi_1, I_{2,\sharp}\xi_2). \quad (2)$$

On a certain subspace of equivalence classes of mm-spaces, this defines a metric, which we call *embedded Wasserstein metric*. For the specific case of Euclidean spaces $X_1$, $X_2$, and $Z$, this reduces to the Wasserstein Procrustes problem (Grave et al., 2019) and to the "embedded Wasserstein metric" in (Salmona et al., 2024). We adopt this name.

**Proposition 3.1.** EW *defines a metric on the subset of isomorphic classes* $[(X, d_X, \xi)]$ *for which there exist surjective isomorphism* $I \colon \operatorname{supp} \xi \hookrightarrow Z$.

By the following proposition, the infimum in (2) is attained.

**Proposition 3.2.** *Let* $\mathbb{X}_i$, $i = 1, 2$ *be two mm-spaces and* $(Z, d_Z)$ *a metric space such that* $\operatorname{supp} \xi_i \hookrightarrow Z$ *for* $i = 1, 2$. *Then the infimum in* (2) *is attained.*

While EW relies on appropriate isometries, the following relaxation enables us to handle arbitrary mm-spaces $\mathbb{X}_i$, $i = 1, 2$. For $\lambda > 0$, we consider the following GW penalized unbalanced OT problem

$$\operatorname{EW}_\lambda(\mathbb{X}_1, \mathbb{X}_2) := \inf_{\pi \in \mathcal{P}(Z \times Z)} \Big( \int_{Z \times Z} d_Z^2(z, z')\,\mathrm{d}\pi(z, z')$$

$$+ \lambda \sum_{i=1}^{2} \operatorname{GW}^2(\mathbb{X}_i, (Z, d_Z, P_{i,\sharp}\pi)) \Big)^{\frac{1}{2}}. \quad (3)$$

By penalizing the GW terms, the marginals $P_{i,\sharp}\pi \in \mathcal{P}(Z)$ take a form that enforces $(Z, d_Z, P_{i,\sharp}\pi)$ to be nearly isomorphic to the inputs $\mathbb{X}_i$, $i = 1, 2$. Furthermore, the first term ensures that the marginals $P_{i,\sharp}\pi$, $i = 1, 2$ are close in the Wasserstein distance on $(Z, d_Z)$. The penalization extends EW to non-isomorphic metric spaces by considering minimum-distortion embeddings. By the following proposition, the infimum in (3) is attained.

**Proposition 3.3.** *Let* $\mathbb{X}_i$, $i = 1, 2$ *be two mm-spaces and* $(Z, d_Z)$ *a metric space. Then* (3) *admits a solution.*

As the GW penalization enforces isometry, EW in (2) becomes the limit of $\mathrm{EW}_\lambda$ in (3) if $\lambda$ goes to infinity.

**Proposition 3.4.** *Let $\mathbb{X}_i$, $i = 1, 2$ be two mm-spaces and let $(Z, d_Z)$ be a metric space such that $\mathrm{supp}\,\xi_i \hookrightarrow Z$. Let $(\lambda_n)_{n\in\mathbb{N}}$ be a sequence with $\lambda_n \to \infty$ as $n \to \infty$. Then any sequence $(\pi_n)_{n\in\mathbb{N}}$ of minimizers of $\mathrm{EW}_{\lambda_n}(\mathbb{X}_1, \mathbb{X}_2)$ converges weakly, up to a subsequence, to some $\pi \in \mathcal{P}(Z \times Z)$. There exist isometries $(I_1, I_2)$ realizing $\mathrm{EW}(\mathbb{X}_1, \mathbb{X}_2)$ such that $\pi \in \Pi_\mathrm{o}(I_{1,\sharp}\xi_1, I_{2,\sharp}\xi_2)$.*

If there exist no isometric embeddings $\mathrm{supp}\,\xi_i \hookrightarrow Z$, the limit of $\mathrm{EW}_\lambda$ as $\lambda \to \infty$ yields best possible approximations of $\mathbb{X}_i$ on the given metric space $(Z, d_Z)$ with respect to the GW distance. More precisely, a GW approximation of an mm-space $\mathbb{X}$ on the metric space $(Z, d_Z)$ is a minimizer $\zeta \in \mathcal{P}(Z)$ of

$$\mathrm{GWA}(\mathbb{X}) := \min_{\zeta \in \mathcal{P}(Z)} \mathrm{GW}^2(\mathbb{X}, (Z, d_Z, \zeta)). \quad (4)$$

For discrete mm-spaces, GW approximations are studied in (Clark et al., 2025) and are closely related to MDS.

**Proposition 3.5.** *Let $\mathbb{X}_i$, $i = 1, 2$, be two mm-spaces and let $(Z, d_Z)$ be a metric space. Let $(\lambda_n)_{n\in\mathbb{N}}$ be a sequence with $\lambda_n \to \infty$ as $n \to \infty$. Then any sequence $(\pi_n)_{n\in\mathbb{N}}$ of minimizers of $\mathrm{EW}_{\lambda_n}(\mathbb{X}_1, \mathbb{X}_2)$ converges weakly, up to a subsequence, to $\pi \in \Pi(\zeta_1, \zeta_2)$, where $\zeta_i \in \mathcal{P}(Z)$ is a GW approximation of $\mathbb{X}_i$.*

On the other side, if $\lambda \to 0$, the influence of the Wasserstein term in $\mathrm{EW}_\lambda$ increases. Figuratively, the marginals $P_{1,\sharp}\pi$ and $P_{2,\sharp}\pi$ of the minimizers $\pi$ of $\mathrm{EW}_\lambda$ have to become more similar if $\lambda$ goes to zero. In the limit case, when both marginals coincide, we obtain a fixed-support GW barycenter (Beier et al., 2023). In detail, a fixed-support GW barycenter on the metric space $(Z, d_Z)$ is a minimizer $\zeta \in \mathcal{P}(Z)$ of

$$\mathrm{GWB}(\mathbb{X}_1, \mathbb{X}_2) := \min_{\zeta \in \mathcal{P}(Z)} \sum_{i=1}^{2} \mathrm{GW}^2(\mathbb{X}_i, (Z, d_Z, \zeta)). \quad (5)$$

**Proposition 3.6.** *Let $\mathbb{X}_i$, $i = 1, 2$, be two mm-spaces and let $(Z, d_Z)$ be a metric space. Let $(\lambda_n)_{n\in\mathbb{N}}$ be a sequence with $\lambda_n \to 0$ as $n \to \infty$. Then any sequence $(\pi_n)_{n\in\mathbb{N}}$ of minimizers of $\mathrm{EW}_{\lambda_n}(\mathbb{X}_1, \mathbb{X}_2)$ converges weakly, up to a subsequence, to $(\mathrm{Id}, \mathrm{Id})_{\sharp}\zeta$, where $\zeta \in \mathcal{P}(Z)$ is a fixed-support GW barycenter.*

In the rest of the paper, we skip the integration domains of the integrals for better readability, since they are clear from the context. Then, by definition of the GW distance, $\mathrm{EW}_\lambda$

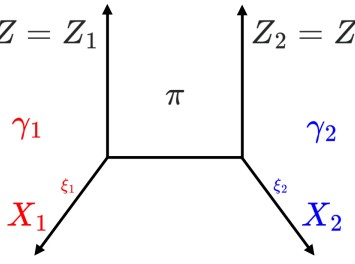

*Figure 3.* Illustration of our multi-marginal transport problem. The colors are in line with Figure 2: quantities with respect to $\mathbb{X}_1$ are red, and quantities related to $\mathbb{X}_2$ are blue.

in (3) can be rewritten as

$$\mathrm{EW}_\lambda^2(\mathbb{X}_1, \mathbb{X}_2) = \inf_{\substack{\pi \in \mathcal{P}(Z \times Z)}} \inf_{\substack{\gamma_i \in \Pi(\xi_i, P_{Z,\sharp}\pi) \\ i=1,2}}$$

$$\int d_Z^2(z, z')\, \mathrm{d}\pi(z, z') + \lambda\big(G_{X_1, Z}(\gamma_1) + G_{X_2, Z}(\gamma_2)\big) \quad (6)$$

or equivalently as

$$\mathrm{EW}_\lambda^2(\mathbb{X}_1, \mathbb{X}_2) = \inf_{\substack{\mu_1, \mu_2 \in \mathcal{P}(Z)}} \inf_{\pi \in \Pi(\mu_1, \mu_2)} \inf_{\substack{\gamma_i \in \Pi(\xi_i, \mu_i) \\ i=1,2}}$$

$$\int d_Z^2(z, z')\, \mathrm{d}\pi(z, z') + \lambda\big(G_{X_1, Z}(\gamma_1) + G_{X_2, Z}(\gamma_2)\big). \quad (7)$$

For computing $\mathrm{EW}_\lambda$, we will rewrite the term by 4-plans. To this end, we use the notation $Z_1 = Z_2 := Z$ and denote the projection onto $X_i, Z_i$ by $P_{X_i}, P_{Z_i}$, respectively. Now we consider 4-plans $\alpha \in \mathcal{P}(X_1, Z_1, Z_2, X_2)$ fulfilling $P_{X_i,\sharp}\alpha = \xi_i$, $i = 1, 2$. Let

$$\pi := P_{Z_1 \times Z_2, \sharp}\alpha \quad \text{and} \quad \gamma_i := P_{X_i \times Z_i, \sharp}\alpha, \ i = 1, 2. \quad (8)$$

Clearly, such plans automatically fulfill

$$P_{Z_i,\sharp}\pi = P_{Z_i,\sharp}\gamma_i, \ i = 1, 2$$

see Figure 3. Thus, (6) can be reformulated as a quadratic, multi-marginal, unbalanced OT problem

$$\mathrm{EW}_\lambda(\mathbb{X}_1, \mathbb{X}_2) = \inf_{\substack{\alpha \in \mathcal{P}(X_1 \times Z_1 \times Z_2 \times X_2) \\ P_{X_i,\sharp}\alpha = \xi_i, i=1,2}} F_\lambda(\alpha)^{\frac{1}{2}} \quad (9)$$

with the quadratic objective

$$F_\lambda(\alpha) := \iint \frac{1}{2}\big(d_Z^2(z_1, z_2) + d_Z^2(z_1', z_2')\big)$$

$$+ \lambda \sum_{i=1}^{2}(d_{X_i}(x_i, x_i') - d_Z(z_i, z_i'))^2$$

$$\times\, \mathrm{d}\alpha(x_1, z_1, z_2, x_2)\, \mathrm{d}\alpha(x_1', z_1', z_2', x_2').$$

We summarize our findings in the following proposition.

**Proposition 3.7.** *If $\alpha$ solves (9), then its projections (8) are solutions of (6) and in particular $\pi$ is a solution of (3). Conversely, any solution of (3) can be expressed this way.*

In Appendix B, we illustrate relations between the Wasserstein distance, GW, EW and $EW_\lambda$ by numerical examples.

## 4. Bi-Convex Relaxation

At its core, the computation of $EW_\lambda$ in (9) requires the solution of a quadratic optimization problem. Similar formulations appear, for instance, in the computation of the GW distance (Peyré et al., 2016; Séjourné et al., 2021), in the multi-marginal GW setting (Beier et al., 2023), and in CO-OT (Vayer et al., 2020b). All of these quadratic OT problem have in common that they can be numerically solved using a block-coordinate descent on their bi-convex relaxations. In the following, we adapt this approach to our multi-marginal, unbalanced transport problem (9).

For this, we decouple the minimization with respect to the inner and outer transport plan $\gamma$ in the double integral of (9). More precisely, denoting the inner plan by $\alpha_1$ and the outer plan by $\alpha_2$, we consider the bi-convex relaxation

$$\inf_{\substack{\alpha_1,\alpha_2\in\mathcal{P}(X_1\times Z_1\times Z_2\times X_2)\\(P_{X_i})_\sharp\alpha_k=\xi_i,\ i,k=1,2}}\mathcal{F}_\lambda(\alpha_1,\alpha_2)\quad(10)$$

with the bilinear objective

$$\mathcal{F}_\lambda(\alpha_1,\alpha_2):=\iint\frac{1}{2}\big(d_Z^2(z_1,z_2)+d_Z^2(z_1',z_2')\big)$$
$$+\lambda\sum_{i=1}^2(d_{X_i}(x_i,x_i')-d_Z(z_i,z_i'))^2$$
$$\times\mathrm{d}\alpha_1(x_1,z_1,z_2,x_2)\,\mathrm{d}\alpha_2(x_1',z_1',z_2',x_2').$$

By construction, the minimizers of (10) constitute a lower bound to the original, quadratic problem (9). Moreover, every bi-convex minimizer of the form $\alpha_1=\alpha_2$ yields a minimizer of the original problem.

In order to find a numerical solution, we apply block-coordinate descent, which consists of alternatively fixing $\alpha_1$ and $\alpha_2$ in (10) and minimizing with respect to the other argument. Fixing $\alpha_1$, it remains to solve the (linear) multi-marginal, unbalanced OT problem

$$\inf_{\substack{P_{X_i,\sharp}\alpha_2=\xi_i\\i=1,2}}\int c_1(x_1',z_1',z_2',x_2')\,\mathrm{d}\alpha_2(x_1',z_1',z_2',x_2')\quad(11)$$

where the effective cost is given by

$$c_1(x_1',z_1',z_2',x_2'):=\frac{1}{2}d_Z^2(z_1',z_2')+\lambda\sum_{i=1}^2$$
$$\int\big(d_{X_i}(x_i,x_i')-d_Z(z_i,z_i')\big)^2\,\mathrm{d}\alpha_1(x_1,z_1,z_2,x_2).\quad(12)$$

The cost function additively decouples into three partial cost functions solely depending on two coordinates of $c_1$. Unbalanced, multi-marginal OT formulations of this kind are treated in (Beier et al., 2022), where, relying on an entropic regularization, a multi-marginal Sinkhorn scheme is proposed. Mathematically, this means that, for a regularization parameter $\varepsilon>0$, we approximate the minimizer of (11) by

$$\inf_{\substack{P_{X_i,\sharp}\alpha_2=\xi_i\\i=1,2}}\int c_1\,\mathrm{d}\alpha_2+\varepsilon\,\mathrm{KL}(\alpha_2,\upsilon),$$

where $\upsilon$ denotes the uniform measure on $X_1\times Z_1\times Z_2\times X_2$ and KL the Kullback–Leibler divergence. More precisely, with the Radon–Nikodým derivative $\mathrm{d}\gamma/\mathrm{d}\upsilon$, the KL divergence is given by $\mathrm{KL}(\gamma,\upsilon):=\int\log(\mathrm{d}\gamma/\mathrm{d}\upsilon)\,\mathrm{d}\upsilon$ if $\gamma\ll\upsilon$ and $\mathrm{KL}(\gamma,\upsilon):=\infty$ otherwise. In total, we may approximate the minimizer of the quadratic, unbalanced, multi-marginal OT problem (9) by applying Algorithm 1.

---

**Algorithm 1** Computation of $EW_\lambda$

---

**input** $\mathbb{X}_i=(X_i,d_{X_i},\xi_i)$, $i=1,2$.      (mm-spaces)
**input** $(Z,d_Z)$      (finite metric space)
**input** $\lambda>0$      (penalization parameter)
**input** $\varepsilon>0$      (regularization parameter)
1: Initialize $\alpha_1=\alpha_2:=\xi_1\otimes\mu\otimes\mu\otimes\xi_2$
    where $\mu$ is the uniform measure on $(Z,d_Z)$
2: **while** not converged **do**
3:     Compute $c_1$ as in (12)
4:     Update $\alpha_2$ using the multi-marginal Sinkhorn scheme in (Beier et al., 2022) with $c_1$
5:     Compute $c_2$ analogous to (12)
6:     Update $\alpha_1$ using the multi-marginal Sinkhorn scheme in (Beier et al., 2022) with $c_2$
7: **end while**
**output** $\alpha_1$ or $\alpha_2$

---

## 5. Discrete $EW_\lambda$ and JMDS

The distance EW in (2) generalizes Wasserstein Procrustes (Grave et al., 2019) to non-Euclidean spaces, while $EW_\lambda$ in (3) is closely related to the JMDS model (Chen et al., 2023). In this section, we explain the relation.

We consider the discrete case, where

$$X_1:=\{x_1^1,\ldots,x_1^{n_1}\}\quad\text{and}\quad X_2:=\{x_2^1,\ldots,x_2^{n_2}\}$$

are point sets equipped with dissimilarity distances $d_{X_1}$ and $d_{X_2}$ as well as measures

$$\xi_1:=\sum_{j=1}^{n_1}\xi_1^j\delta_{x_1^j},\quad\xi_2:=\sum_{j=1}^{n_2}\xi_2^j\delta_{x_2^j}.$$

**Discrete $EW_\lambda$** fixes a discrete embedding space

$$Z_1=Z_2=Z:=\{z^1,\ldots,z^m\}.$$

with metric $d_Z$. Accordingly, we use measures with fixed supports

$$\mu_i = \sum_{j=1}^{m} \mu_i^j \delta_{z^j}, \; \gamma_i = \sum_{j,k=1}^{n_i,m} \gamma_i^{j,k} \delta_{x_i^j, z^k}, \; \pi = \sum_{j,k=1}^{m} \pi_{j,k} \delta_{z^j, z^k},$$

where $i = 1, 2$. Since the supports of the measures are fixed, we can restrict ourselves to the weight matrices $\boldsymbol{\mu}_i \in \Delta_m$, $\boldsymbol{\xi}_i \in \Delta_{n_i}$, $\boldsymbol{\pi} \in \Pi(\boldsymbol{\mu}_1, \boldsymbol{\mu}_2)$ and $\boldsymbol{\gamma}_i \in \Pi(\boldsymbol{\xi}_i, \boldsymbol{\mu}_i)$, $i = 1, 2$ in the corresponding probability simplices $\Delta$. Then $\mathrm{EW}_\lambda$ in (7) becomes the discrete minimization problem

$$\mathrm{EW}_\lambda(X_1, X_2) = \min_{\substack{\boldsymbol{\mu}_i \in \Delta_m \\ i=1,2}} \min_{\boldsymbol{\pi} \in \Pi(\boldsymbol{\mu}_1, \boldsymbol{\mu}_2)} \min_{\substack{\boldsymbol{\gamma}_i \in \Pi(\boldsymbol{\xi}_i, \boldsymbol{\mu}_i) \\ i=1,2}}$$
$$\langle \boldsymbol{\pi}, \boldsymbol{D}_{Z,Z}^2 \rangle + \lambda \big( G_{X_1, Z}(\boldsymbol{\gamma}_1) + G_{X_2, Z}(\boldsymbol{\gamma}_2) \big),$$

where $\boldsymbol{D}_{Z,Z}^2 := \big(d_Z^2(z^j, z^k)\big)_{j,k=1}^m$ and, for $i = 1, 2$,

$$G_{X_i, Z}(\boldsymbol{\gamma}_i)$$
$$:= \sum_{j,k=1}^{n_i,m} \sum_{r,s=1}^{n_i,m} \gamma_i^{s,k} \gamma_i^{j,k} \big( d_{X_1}(x_i^j, x_i^r) - d_Z(z^k, z^s) \big)^2.$$

**JMDS** aims to find point sets

$$\mathcal{Z}_1 := \{z_1^1, \dots, z_1^{n_1}\} \subset \mathbb{R}^d, \quad \mathcal{Z}_2 := \{z_2^1, \dots, z_2^{n_2}\} \subset \mathbb{R}^d$$

such that the dissimilarity relations in $X_i$ are approximately preserved in $\mathcal{Z}_i$, $i = 1, 2$, while ensuring that the intermediate points are optimally aligned. Here, $\mathcal{Z}_i$, $i = 1, 2$ are exclusively in $\mathbb{R}^d$ with the Euclidean metric $d(x, y) = \|x - y\|$. Instead of measures with fixed support, we use

$$\mu_i := \sum_{j=1}^{n_i} \frac{1}{n_i} \delta_{z_i^j}, \; \pi = \sum_{j,k=1}^{m} \pi_{j,k} \delta_{z_1^j, z_1^k},$$

as well as

$$\gamma_i := \sum_{j,k=1}^{n_i} \gamma_i^{j,k} \delta_{x_i^j, z_i^k} \quad \text{with} \quad \gamma_i^{j,k} := \begin{cases} \frac{1}{n_i} & j = k, \\ 0 & j \neq k, \end{cases}$$

and will optimize over the supports $\mathcal{Z}_i$, $i = 1, 2$. Then (7) becomes

$$\mathrm{JMDS}_\lambda(X_1, X_2) := \min_{\mathcal{Z}_1, \mathcal{Z}_2} \min_{\boldsymbol{\pi} \in \Pi(\frac{1}{n_1} \mathbf{1}_{n_1}, \frac{1}{n_2} \mathbf{1}_{n_2})}$$
$$\langle \boldsymbol{\pi}, \boldsymbol{D}_{\mathcal{Z}_1, \mathcal{Z}_2}^2 \rangle + \lambda \big( G_1(\mathcal{Z}_1) + G_2(\mathcal{Z}_2) \big), \quad (13)$$

where $\boldsymbol{D}_{\mathcal{Z}_1, \mathcal{Z}_2}^2 := \big(d^2(z_1^j, z_2^k)\big)_{j,k=1}^{n_1, n_2}$ and, for $i = 1, 2$,

$$G_i(\mathcal{Z}_i) := \sum_{j,k=1}^{n_i} \sum_{r,s=1}^{n_i} \gamma_i^{j,k} \gamma_i^{r,s} \big( d_{X_i}(x_i^j, x_i^r) - d(z_i^k, z_i^s) \big)^2$$

$$= \sum_{j,k=1}^{n_i} \frac{1}{n_i^2} \big( d_{X_i}(x_i^j, x_i^k) - d(z_i^j, z_i^k) \big)^2.$$

This is exactly the functional proposed as *JMDS* in (Chen et al., 2023). Note that the *Wasserstein Procrustes model* is given by

$$\min_{\boldsymbol{\pi} \in \Pi(\frac{1}{n_1} \mathbf{1}_{n_1}, \frac{1}{n_2} \mathbf{1}_{n_2})} \min_{Q \in \mathrm{SO}(d)} \langle \boldsymbol{\pi}, \boldsymbol{D}_{Q\mathcal{Z}_1, \mathcal{Z}_2}^2 \rangle.$$

This is exactly the first summand in $\mathrm{JMDS}_\lambda$, where the minimization over the special orthogonal group $\mathrm{SO}(d)$ can be skipped in (13) since $G_i(\mathcal{Z}_i)$, $i = 1, 2$, are invariant under orthogonal transforms.

To summarize: 1. Our discrete $\mathrm{EW}_\lambda$ fixes the support of the marginals $\mu_i$ in (7) which results in the optimization over the weights, where JMSD fixes the weights of the $\mu_i$ and optimizes over the supports. 2. Fixing the support has the advantage that we can work with arbitrary metric spaces $(Z, d_Z)$, while the "free support" approach of JMDS is restricted to the Euclidean space. 3. The minimization problems have to be tackled by completely different optimization algorithms, namely an unbalanced multimarginal Sinkhorn algorithm for the block-coordinate descent in Algorithm 1 for $\mathrm{EW}_\lambda$ and the so-called SMACOF method combined with Wasserstein Procrustes minimizations (Chen et al., 2023) for JMDS.

## 6. Numerical Results

Next, we provide several proof-of-concept examples[1]. Additional parameter ablations are described in Supplement C.

### 6.1. Joint Embedding of 3d Shapes

In the first example, we exploit $\mathrm{EW}_\lambda$ to align and embed 3d shapes—the surfaces of objects in $\mathbb{R}^3$—into a joint space $(Z, d_Z)$. For this, we interpret a 3d shape as mm-space $\mathbb{X} := (X, d_X, \xi)$, where $X$ is the surface, $d_X$ is the surface (or geodesic) distance, and $\xi$ is the uniform measure. Practically, $X$ is parametrized by the vertices of a triangular mesh, $d_X$ is approximated using Dijkstra's algorithm (Dijkstra, 1959) on the corresponding graph, and $\xi$ is chosen as the discrete uniform measure. For the joint embedding of two given (discrete) shapes $\mathbb{X}_i$, $i = 1, 2$, we compute the 4-plan $\alpha$ in (9) using the discretization in Section 5 and Algorithm 1. Since the relaxation behind $\mathrm{EW}_\lambda$ does not yield an isometry, but only a transport plan $\gamma_i = P_{X_i \times Z_i, \sharp} \alpha$, we visualize the computed relaxed embeddings by the marginals $P_{Z_i, \sharp} \gamma_i = P_{Z_i, \sharp} \alpha$.

**Bended Rectangles** We start by embedding an S-bended rectangle and a Swiss roll (with and without a hole) into $\mathbb{R}^2$. Intuitively, we expect that the surfaces without holes are unrolled by the relaxed embedding behind $\mathrm{EW}_\lambda$, since there

---

[1] https://github.com/MoePien/RelaxedEmbeddedWasserstein

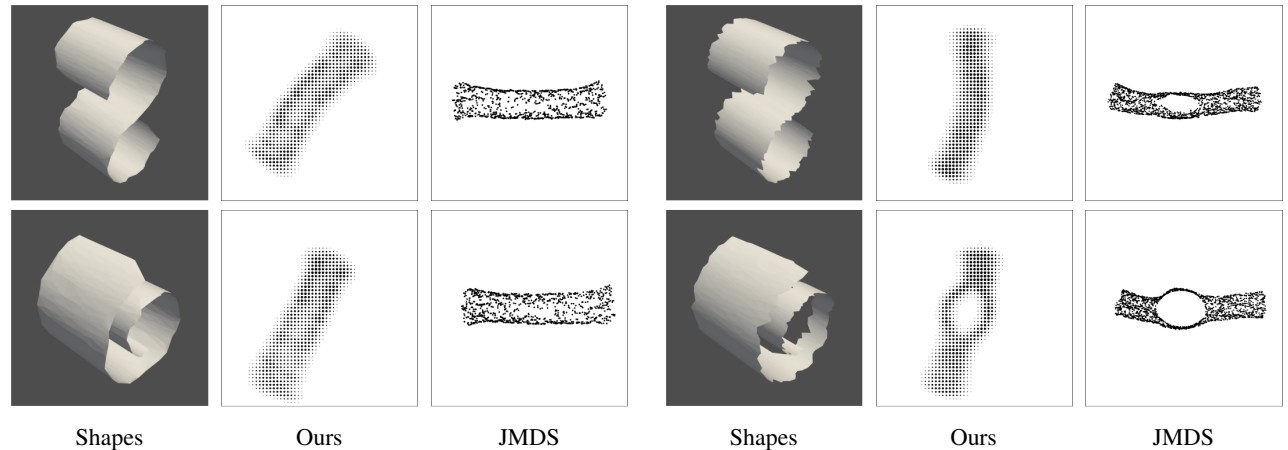

(a) S-bended rectangle and Swiss roll (without hole).  (b) S-bended rectangle and Swiss roll with hole.

*Figure 4.* Embedding and alignment of an S-bended rectangle and a Swiss roll into $\mathbb{R}^2$. For our method, we visualize the marginals $P_{Z_i,\sharp}\alpha$ of the computed 4-plan $\alpha$ in (9) and compare them with JMDS. In the second example (b), JMDS produces an unexpected hole when embedding the S-bended surface.

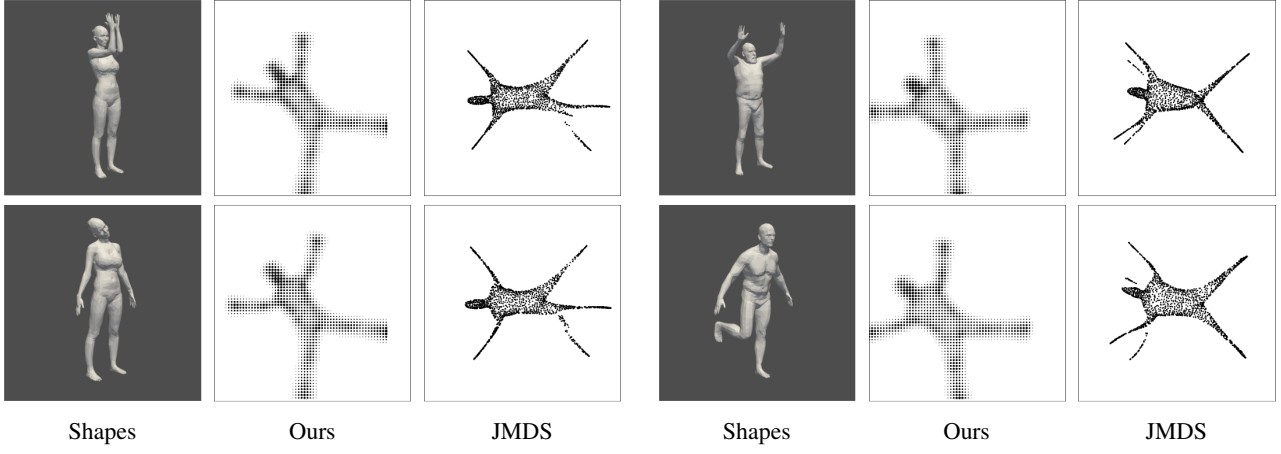

(a) Same human shape in different poses.  (b) Distinct human shapes in different poses.

*Figure 5.* Embedding and alignment of human shapes from the FAUST dataset into $\mathbb{R}^2$. For our method, we visualize the marginals $P_{Z_i,\sharp}\alpha$ of the computed 4-plan $\alpha$ in (9) and compare them with JMDS. Here JMDS tends to split some of the extremities.

actually exists isometric embeddings. For the discretization, we choose $Z$ as an equispaced $50\times50$ grid on $[0, 1.3]^2 \subset \mathbb{R}^2$ and $d_Z$ as the corresponding Euclidean distance and apply Algorithm 1 with $\lambda = 100$ and $\varepsilon = 10^{-3}$. In Figure 4, the relaxed embeddings are visualized by $P_{Z_i,\sharp}\alpha$, where the point sizes represent the underlying probability masses/weights. As comparison, we also show the results of JMDS with $\lambda = 10$ and $\varepsilon = 10^{-3}$ for the incorporated regularized OT problem. Both methods produce representations that unroll the shapes and succeed in aligning the embeddings. JMDS, however, produces an unexpected hole during the alignment of the S-bended shape and the Swiss role with hole.

**Human Shapes** Since we are not limited to isometries, we next consider an experiment where isometric embeddings

are not possible. More precisely, we embed human shapes from the FAUST dataset (Bogo et al., 2014) into $\mathbb{R}^2$. For this, we choose $(Z, d_Z)$ as an equispaced $60\times60$ grid on $[0, 1.3]^2 \subset \mathbb{R}^2$ and apply Algorithm 1 with $\lambda = 100$ and $\varepsilon = 4 \cdot 10^{-4}$. The obtained transport-based embeddings are shown in Figure 5. As comparison, we apply JMDS with $\lambda = 10$ and $\varepsilon = 4 \cdot 10^{-4}$. The 2d representations clearly resemble the 3d human shapes, but JMDS splits some of the extremities. Since our method and JMDS can both be derived from the same functional, we additionally compare the achieved objectives, the Wasserstein distances between embeddings, and the GW distances between embeddings and input spaces. The experiment is repeated for each pairwise combination of the first ten shapes from the FAUST dataset and different $\lambda$. For comparison, the shapes are

| | GW$^2$ | W$^2$ |
|---|---|---|
| Ours ($\lambda = 1$) | 0.0057±0.004 | 0.017±0.007 |
| JMDS ($\lambda = 1$) | 0.0071±0.007 | 0.029±0.010 |
| Ours ($\lambda = 10$) | 0.0048±0.004 | 0.027±0.016 |
| JMDS ($\lambda = 10$) | 0.0057±0.005 | 0.046±0.023 |

*Table 1.* Mean and standard deviation of GW distance (between input shapes and embeddings) and of Wasserstein distance (between both embeddings) for joint 2d embeddings of the FAUST dataset via our method and JMDS.

downsampled to 35 vertices, and both methods are regularized with $\varepsilon = 0.001$. For our method, we set $Z$ according to a uniform 30×30 grid on $[0, 1.3]^2$. Results are shown in Table 1. In total, we here achieve better joint embeddings than JMDS both in terms of GW and Wasserstein.

**Spherical and Toroidal Embeddings** Finally, we consider the alignment of spherical and toroidal subsets, i.e., the joint embedding into a non-Euclidean space. More precisely, we consider the 3d shapes in the introductory Figure 1. As target space $(Z, d_Z)$, we choose a 30×30 grid on the canonical parametrization of the sphere $\mathbb{S}(2)$ and the torus $\mathbb{T}(2) = \mathbb{S}(1) \times \mathbb{S}(1)$ equipped with the corresponding geodesic distance. Using $\lambda = 10^3$ and $\varepsilon = 10^{-3}$ for Algorithm 1, we embed the spherical rectangle and a spherical cap, both with a hole, onto the sphere, and the half-torus and toroidal triangle onto the torus. The relaxed embeddings are shown in Figure 1, where the color encodes the mass of the computed marginals. Up to a smoothing due to the incorporated entropic regularization, the transport plans $\gamma_i = P_{X_i \times Z_i, \sharp} \alpha$ correspond to the expected isometries.

### 6.2. Alignment of Feature Spaces

In the next example, we use our method to align different feature spaces occurring in real-world data. More precisely, we consider the genetyping—the determination of the corresponding cell class—of single cells from various measured modalities. Multimodal technologies as proposed in (Cheow et al., 2016; Chen et al., 2019) are, however, uncommon such that there is a rising interest in embedding the corresponding features into a joint feature space (Demetci et al., 2022; Liu et al., 2019; Chen et al., 2023). Following the experiments of (Demetci et al., 2022) for $n$ single cells, our first aim is to assign the recorded features $X_1 \coloneqq \{x_1^1, \ldots, x_1^n\} \subset \mathbb{R}^{d_1}$ from one modality to the features $X_2 \coloneqq \{x_2^1, \ldots, x_2^n\} \subset \mathbb{R}^{d_2}$ of another modality, i.e., to recover the underlying one-to-one correspondence between $x_1^j$ and $x_2^j$. Our second goal consists in transferring a classifier from $X_1$ to $X_2$ in an unsupervised manner.

To achieve both goals, we apply the following methodology:
1. We equip the recorded, uncorrelated features $X_1 \subset \mathbb{R}^{d_1}$

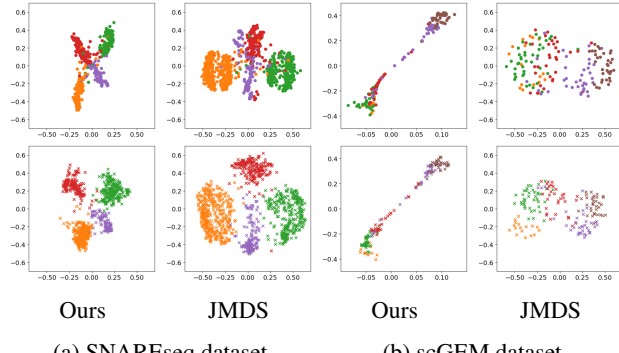

(a) SNAREseq dataset      (b) scGEM dataset

*Figure 6.* Joint embedding of two feature spaces into $\mathbb{R}^2$ using our method and JMDS (Top: first feature space, Bottom: second feature space). Both methods align the color-coded classes.

and $X_2 \subset \mathbb{R}^{d_2}$ with specific distances $d_{X_i}$ estimated with the unsupervised SCOT routine (Demetci et al., 2022).
2. We equip the constructed metric spaces with the uniform measure to obtain the mm-spaces $\mathbb{X}_i \coloneqq (X_i, d_{X_i}, \xi_i)$.
3. Fixing a 20×20 grid $Z \coloneqq \{z^1, \ldots, z^m\} \subset \mathbb{R}^2$ over $[0, 1]^2$ equipped with the Euclidean metric, we apply Algorithm 1 with the discretization in Section 5 to compute the 4-plan $\alpha$ in (9).
4. Based on the transport plans $\gamma_i \coloneqq P_{X_i \times Z_i, \sharp} \alpha$, we pair $x_i^j$ with its barycentric projection $\tilde{x}_i^j \in \mathbb{R}^m$ given by

$$\tilde{x}_i^j \coloneqq \sum_{k=1}^m \gamma_i^{jk} z_k \bigg/ \sum_{k=1}^m \gamma_i^{jk}.$$

For the first aim, the identification of the underlying correspondence, we identify $x_1^j$ with $x_2^k$ whose barycentric projection $\tilde{x}_2^k$ is closest to $\tilde{x}_1^j$. To quantify the quality of the pairing, we rely on the FOSCTTM (fraction of samples closer than the true match) score (Liu et al., 2019). More precisely, FOSCTTM is defined by

$$\frac{1}{n^2} \sum_{j=1}^n \#\Big\{k \in \{1, \ldots, n\} : \|\tilde{x}_1^j - \tilde{x}_2^k\| < \|\tilde{x}_1^j - \tilde{x}_2^j\|\Big\}$$

and ranges from 0 (perfect identification) to 1.

For the second aim, the transfer of a classifier from $X_1$ to $X_2$, we exemplarily consider the k-nearest neighbor (KNN) method. More precisely, we use the barycentric projections $\{\tilde{x}_1^1, \ldots, \tilde{x}_1^n\}$ and the corresponding gene type labels of the first modality to classify a feature $x_2^k$ (more exact $\tilde{x}_2^k$) from the second modality, where we consider the closest five neighbors. Here, higher classification accuracies (KNN-Acc) indicate better class alignments.

For the experiments, we employ the publicly available datasets[2] from (Demetci et al., 2022), where SNAREseq

[2] https://rsinghlab.github.io/SCOT/data/

| | SNAREseq | | | scGEM | | |
|---|---|---|---|---|---|---|
| | FOSCTTM↓ | KNN-Acc↑ | Time | FOSCTTM↓ | KNN-Acc↑ | Time |
| EW$_\lambda$ (Ours) | **0.165** | **0.943** | 28.9s | **0.208** | 0.610 | 1.7s |
| JMDS (Chen et al., 2023) | 0.219 | 0.872 | 8.7s | 0.230 | **0.616** | 0.3s |
| SCOT (Demetci et al., 2022) | 0.402 | 0.447 | 76.7s | 0.247 | 0.578 | 17.6s |
| UnionCom (Cao et al., 2020) | 0.436 | 0.380 | 90.6s | 0.392 | 0.465 | 1.5s |

*Table 2.* Comparison of the joint embedding of two feature spaces into $\mathbb{R}^2$ using our method, JMDS, SCOT, and UnionCom.

consists of $d_1 = 19$ and $d_2 = 10$ features of $n = 1047$ single cells, and scGEM of $d_1 = 34$ and $d_2 = 27$ features of $n = 177$ specimens. The computed embedding of $X_i$ into $\mathbb{R}^2$ of our method and JMDS are visualized in Figure 6. For FOSCTTM and KKN-Acc, we additionally compare both methods with the alignment techniques UnionCom (Cao et al., 2020) and SCOT (Demetci et al., 2022) using the same distances on $X_i$. Note that UnionCom relies on GUMA and t-SNE, whereas SCOT relies on GW and MDS. The results are recorded in Table 2. The hyperparameters of all methods are chosen according to a grid search minimizing the FOSCTTM on a 10% validation split, see Supplement D. A further experiment with different feature spaces for MNIST and FashionMNIST is given in Supplement E.

### 6.3. Alignment of Gaussian Mixture Models

In the final example, inspired by (Salmona et al., 2024), we want to align Gaussian mixture models (GMMs). For this, we consider the space of 2d Gaussians $\mathcal{N}_2 = \{\mathcal{N}(\mu, \Sigma) | \mu \in \mathbb{R}^2, \Sigma \in \mathbb{R}^{2 \times 2} \text{ spd.}\}$ equipped with the Wasserstein distance, which yields a non-Euclidean geometry. A GMM can now be interpreted as mm-space $\mathbb{X} = (\mathcal{N}_2, W, \xi)$ with discrete measure $\xi := \sum_{j=1}^{n_1} \xi_j \, \delta_{\mu_j, \Sigma_j}$. For the experiment, we fit two GMMs to affine-transformed 2d datasets using the expectation-maximization algorithm. Here, we employ the datasets "Blobs" and "Moons" from scikit-learn (Pedregosa et al., 2011), see Figure 7. For the alignment of the GMMs, we choose $Z \subset \mathcal{N}_2$, where the means form a $15 \times 15$ grid over $[0, 1]^2$ and we use a coarse grid over suitable matrices. Particularly, we consider matrices

$$\Sigma = r^2 \begin{bmatrix} \sigma_1^2 & \sigma_{12} \\ \sigma_{12} & \sigma_2^2 \end{bmatrix},$$

where we choose $\sigma_1^2, \sigma_2^2 \in \{0.8, 1.\}$ and $\sigma_{12} \in \{-0.2, 0., 0.2\}$. The parameter $r^2 > 0$ corresponds to the mean variance of the considered GMMs. Applying Algorithm 1 to compute $\alpha$ in (9), and considering the marginals $\zeta_i := P_{Z_i, \sharp} \alpha$, which are again discrete measures on $\mathcal{N}_2$, we obtain the aligned GMMs in Figure 7.

### 7. Conclusion

We propose an unbalanced OT framework with GW marginal penalization which enables the joint embedding of two datasets based on pairwise intra-dataset distances.

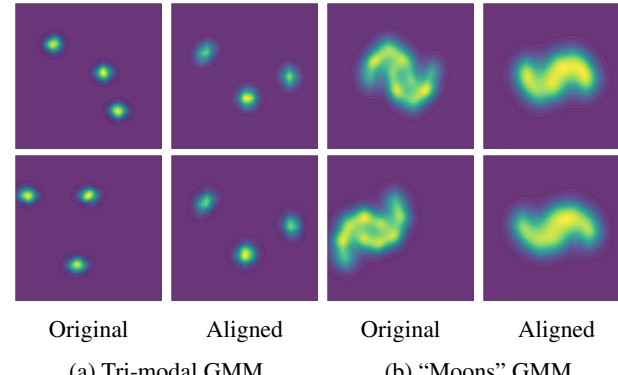

| Original | Aligned | Original | Aligned |
|---|---|---|---|
| (a) Tri-modal GMM | | (b) "Moons" GMM | |

*Figure 7.* Alignment of GMMs with respect to the Wasserstein distance. The considered and computed GMMs are visualized via their density function on $\mathbb{R}^2$.

While our model can handle the aligned transfer into arbitrary metric spaces, it relies on their appropriate definition. In particular, due to computational restrictions, working on a grid in $\mathbb{R}^d$ is restricted to small dimensions $d$. As a future research direction, we are interested in developing a non-Euclidean free-support solver based on existing free-support GW barycenter algorithms (Peyré et al., 2016; Vayer et al., 2020a; Beier & Beinert, 2025). This may combine our EW$_\lambda$ approach with a non-Euclidean version of JMDS.

From a theoretical point of view, the joint embedding using proposed variational formulation may be adapted to handle multiple input spaces. The main issue is hereby the selection of an appropriate generalization of the employed Wasserstein part. A possible choice would be to utilize only the Wasserstein distance between the embeddings of the $k$th and $(k + 1)$st embedding. Since the employed multi-marginal Sinkhorn method linearly scales in the number of spaces, we expect that the resulting method remains numerically tractable for small numbers of spaces. The actual choice of the model and the numerical implementation of a multiple space variant, however, requires further studies.

### Acknowledgements

M.P. acknowledges funding from the German Research Foundation (DFG, GRK2260 BIOQIC project 289347353).

### Impact Statement

This paper presents work whose goal is to advance the field of Machine Learning. There are many potential societal consequences of our work, none of which we feel must be specifically highlighted here.

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

# A. Proofs

**Proof of Proposition 3.1** The main idea of the proof follows the argumentation for the original embedded Wasserstein distance on Euclidean spaces in (Salmona et al., 2024), which we adapt to arbitrary metric spaces. The definiteness follows directly from the definition of the isomorphic equivalence classes, the isometries in EW, and the definiteness of the Wasserstein distance. The symmetry is inherited from the Wasserstein distance too. It remains to show the triangle inequality. Without loss of generality, we take representatives $\mathbb{X}_i := (Z, d_Z, \xi_i)$, $i = 1, \ldots, 3$, such that $\xi_i \in \mathcal{P}(Z)$ possess full support. Since the inverse and composition of isometries are isometry, we have

$$\text{EW}(\mathbb{X}_1, \mathbb{X}_2) = \inf_{I_i : Z \hookrightarrow Z} \text{W}(I_{1,\sharp}\xi_1, I_{2,\sharp}\xi_2) = \inf_{I_i : Z \hookrightarrow Z} \text{W}(\xi_1, (I_1^{-1} \circ I_2)_\sharp \xi_2) = \inf_{I : Z \hookrightarrow Z} \text{W}(\xi_1, I_\sharp \xi_2).$$

Using the triangle inequality of the Wasserstein distance, we obtain

$$\text{EW}(\mathbb{X}_1, \mathbb{X}_2) = \inf_{I : Z \hookrightarrow Z} \text{W}(\xi_1, I_\sharp \xi_2) \leq \inf_{I, J : Z \hookrightarrow Z} \text{W}(\xi_1, J_\sharp \xi_3) + \text{W}(J_\sharp \xi_3, I_\sharp \xi_2)$$

$$= \inf_{J : Z \hookrightarrow Z} \left( \text{W}(\xi_1, J_\sharp \xi_3) + \inf_{I : Z \hookrightarrow Z} \text{W}(J_\sharp \xi_3, I_\sharp \xi_2) \right) = \inf_{J : Z \hookrightarrow Z} \left( \text{W}(\xi_1, J_\sharp \xi_3) + \inf_{I : Z \hookrightarrow Z} \text{W}(\xi_3, (J^{-1} \circ I)_\sharp \xi_2) \right)$$

$$= \inf_{J : Z \hookrightarrow Z} \text{W}(\xi_1, J_\sharp \xi_3) + \inf_{K : Z \hookrightarrow Z} \text{W}(\xi_3, K_\sharp \xi_2) = \text{EW}(\mathbb{X}_1, \mathbb{X}_3) + \text{EW}(\mathbb{X}_3, \mathbb{X}_2). \qquad \square$$

**Proof of Proposition 3.2** Let $X \in \{X_i : i = 1, 2\}$. We equip the space of continuous functions $C(X, Z)$ with the topology of uniform convergence $d(f, g) := \max_{x \in X} d_Z(f(x), g(x))$.

1. First, we we show that

$$\text{Iso}(X, Z) := \{I : X \hookrightarrow Z\} \subset C(X, Z)$$

is compact. Since $(Z, d_Z)$ is compact, we see immediately that $\text{Iso}(X, Z)$ is pointwise bounded. Moreover, since all functions in $\text{Iso}(X, Z)$ are isometries the set is equicontinuous. By the Theorem of Arzelà–Ascoli (Munkres, 2000) (Thm. 47.1), this gives the relative compactness of Iso. Further, $\text{Iso}(X, Z) \subset C(X, Z)$ is closed and thus compact, since we have for any sequence of isometies $(I_n)_n$ from $X$ to $Z$ converging to some $I \in C(X, Z)$ that

$$d_X(x, x') = d_Z(I_n(x), I_n(x')) \to d_Z(I(x), I(x'))$$

as $n \to \infty$, so that $I \in \text{Iso}(X, Z)$. Then also $\text{Iso}(X_1, Z) \times \text{Iso}(X_2, Z)$ is compact.
2. Next, we show that the functional

$$F(I_1, I_2) := \text{W}(I_{1,\sharp}\xi_1, I_{2,\sharp}\xi_2)$$

in (2) is lower semi-continuous on $\text{Iso}(X_1, Z) \times \text{Iso}(X_2, Z)$. For $i = 1, 2$, let $(I_{i,n})_{n \in \mathbb{N}} \subset \text{Iso}(X_i, Z)$ and $I_i \in \text{Iso}(X_1, Z)$ with $I_{i,n} \to I_i$ as $n \to \infty$. Then, for all $\varphi \in C(Z, \mathbb{R})$, it holds by the dominated convergence theorem that

$$\int_Z \varphi \, \mathrm{d}(I_{i,n})_\sharp \xi_i = \int_X \varphi \circ I_{i,n} \, \mathrm{d}\xi_i$$

$$\to \int_{X_i} \varphi \circ I_i \, \mathrm{d}\xi_i = \int_Z \varphi \, \mathrm{d}I_{i,\sharp}\xi_i \ \text{ as } \ n \to \infty.$$

Thus we obtain the weak convergence $(I_{i,n})_\sharp \xi_i \rightharpoonup I_{i,\sharp}\xi_i$. Finally, due to the joint weak lower semi-continuity of the Wasserstein distance (Ambrosio et al., 2005) (Lemma. 7.1.4), we get the desired lower semi-continuity of $F$.

Finally, application of the Weierstrass theorem yields the assertion. $\qquad \square$

**Proof of Proposition 3.3** Since $\mathcal{P}(Z \times Z)$ is weakly compact, the existence of a minimizer follows by Weierstrass' theorem, see, e.g., (Mordukhovich & Nam, 2022) (Thm. 2.164), once we can show that the objective function

$$F(\pi) := \int_{Z \times Z} d_Z^2(z, z') \, \mathrm{d}\pi(z, z') + \lambda \sum_{i=1}^2 \text{GW}^2(\mathbb{X}_i, (Z, d_Z, P_{i,\sharp}\pi))$$

in (3) is weakly lower semi-continuous.

Let $(\pi_n)_{n\in\mathbb{N}} \subset \mathcal{P}(Z \times Z)$ and $\pi \in \mathcal{P}(Z \times Z)$ be such that $\pi_n \rightharpoonup \pi$ as $n \to \infty$. We handle the terms of $F$ separately. Since $d_Z^2$ is continuous in both arguments, we obtain by definition of the weak convergence that

$$\int_{Z\times Z} d_Z^2(z,z') \, \mathrm{d}\pi_n(z,z') \to \int_{Z\times Z} d_Z^2(z,z') \, \mathrm{d}\pi(z,z')$$

as $n \to \infty$. We turn to the GW terms. For $i = 1, 2$, set

$$\mu_{i,n} := P_{i,\sharp}\pi_n \quad \text{and} \quad \mu_i := P_{i,\sharp}\pi.$$

Then the weak lower semi-continuity of the marginal projection operators yields $\mu_{i,n} \rightharpoonup \mu_i$ as $n \to \infty$. Now let $\gamma_{i,n} \in \Pi(\xi_i, \mu_{i,n})$ be a solution of $\mathrm{GW}(\mathbb{X}_i, (Z, d_Z, \mu_{i,n}))$, $n \in \mathbb{N}$. As $(\gamma_{i,n})_{n\in\mathbb{N}} \subset \mathcal{P}(X_i \times Z)$ and the latter is weakly compact, we can choose a converging subsequence again denoted by $(\gamma i, n)_{n\in\mathbb{N}}$, such that $\gamma_{i,n} \rightharpoonup \gamma_i \in \mathcal{P}(X_i \times Z)$. Using again the weak continuity of the marginal projection, we get $\gamma \in \Pi(\xi_i, \mu_i)$. Therefore, it holds

$$\mathrm{GW}^2(\mathbb{X}_i, (Z, d_Z, P_{i,\sharp}\pi)) = \mathrm{GW}^2(\mathbb{X}_i, (Z, d_Z, \mu_i)) \leq \iint_{(X_i\times Z)^2} (d_{X_i}(x_i, x_i') - d_Z(z,z'))^2 \, \mathrm{d}\gamma(x_i, z) \, \mathrm{d}\gamma(x_i', z')$$

and by continuity of the integrand and the fact that $(\gamma_{i,n} \otimes \gamma_{i,n}) \rightharpoonup (\gamma_i \otimes \gamma_i)$, see, e.g., (Bogachev, 2018) (Prop. 2.7.8) further

$$\mathrm{GW}^2(\mathbb{X}_i, (Z, d_Z, P_{i,\sharp}\pi)) \leq \lim_{n\to\infty} \iint_{(X_i\times Z)^2} (d_{X_i}(x_i, x_i') - d_Z(z,z'))^2 \, \mathrm{d}\gamma_{i,n}(x_i, z) \, \mathrm{d}\gamma_{i,n}(x_i', z')$$
$$= \lim_{n\to\infty} \mathrm{GW}^2(\mathbb{X}_i, (Z, d_Z, \mu_{i,n})) = \lim_{n\to\infty} \mathrm{GW}^2(\mathbb{X}_i, (Z, d_Z, P_{i,\sharp}\pi_n)).$$

This gives lower semi-continuity of the functional of $F$. $\qquad\square$

**Proof of Proposition 3.4** 1. First, we show that

$$\mathrm{EW}_\lambda(\mathbb{X}_1, \mathbb{X}_2) \leq \mathrm{EW}(\mathbb{X}_1, \mathbb{X}_2), \qquad \lambda > 0. \tag{14}$$

To this end, let $I_i : X_i \hookrightarrow Z_i$ be isometries realizing $\mathrm{EW}(\mathbb{X}_1, \mathbb{X}_2)$, i.e.,

$$\mathrm{EW}(\mathbb{X}_1, \mathbb{X}_2) = \mathrm{W}(I_{1,\sharp}\xi_1, I_{2,\sharp}\xi_2).$$

By definition of the GW distance, we have $\mathrm{GW}(\mathbb{X}_i, (Z_i, d_{Z_i}, I_{i,\sharp}\xi_i)) = 0$, $i = 1, 2$. Then we obtain for $\tilde{\pi} \in \Pi_0(I_{1,\sharp}\xi_1, I_{2,\sharp}\xi_2)$ that

$$\mathrm{EW}_\lambda(\mathbb{X}_1, \mathbb{X}_2) \leq \left( \int_{Z_1\times Z_2} d_Z^2(z,z') \, \mathrm{d}\tilde{\pi}(z,z') \right)^{\frac{1}{2}} = \mathrm{W}(I_{1,\sharp}\xi_1, I_{2,\sharp}\xi_2) = \mathrm{EW}_{(Z,d_Z)}(\mathbb{X}_1, \mathbb{X}_2).$$

2. Now let $(\lambda_n)_{n\in\mathbb{N}}$ with $\lambda_n \to \infty$ as $n \to \infty$, and let $\pi_n \in \mathcal{P}(Z \times Z)$ realize $\mathrm{EW}_{\lambda_n}(\mathbb{X}_1, \mathbb{X}_2)$. Then we conclude by (14) that

$$\mathrm{EW}(\mathbb{X}_1, \mathbb{X}_2) \geq \mathrm{EW}_{\lambda_n}(\mathbb{X}_1, \mathbb{X}_2) = \int_{Z\times Z} d_Z(z,z') \, \mathrm{d}\pi_n(z,z') + \lambda_n \sum_{i=1}^2 \mathrm{GW}(\mathbb{X}_i, (Z, d_Z, P_{i,\sharp}\pi_n))$$
$$\geq \lambda_n \sum_{i=1}^2 \mathrm{GW}(\mathbb{X}_i, (Z, d_Z, (P_i)_{\#}\pi_n)).$$

Hence, we obtain for $i = 1, 2$ that

$$\mathrm{GW}(\mathbb{X}_i, (Z, d_Z, (P_i)_{\#}\pi_n)) \to 0 \quad \text{as} \quad n \to \infty.$$

Since $(\pi_n)_{n\in\mathbb{N}}$ is contained in the weakly compact set $\mathcal{P}(Z \times Z)$, there exists a subsequence (denoted in the same way) which weakly converges to some $\pi \in \mathcal{P}(Z \times Z)$. Then also $P_{i,\sharp}\pi_n$ converges weakly to $P_{i,\sharp}\pi$, $i = 1, 2$ and we obtain as in the proof of (3.3), that

$$\mathrm{GW}(\mathbb{X}_i, (Z, d_Z, P_{i,\sharp}\pi)) \leq \lim_{n\to\infty} \mathrm{GW}(\mathbb{X}_i, (Z, d_Z, P_{i,\sharp}\pi_n)) = 0.$$

Thus, there exist isometries $I_i : X_i \hookrightarrow Z$ with $P_{i,\sharp}\pi = I_{i,\sharp}\xi_i$, $i = 1, 2$, i.e., $\pi \in \Pi(I_{1,\sharp}\xi_1, I_{2,\sharp}\xi_2)$. Finally, we obtain for any fixed $\lambda > 0$ that

$$
\begin{aligned}
\mathrm{EW}(\mathbb{X}_1, \mathbb{X}_2) &\leq \Big( \int_{Z_1 \times Z_2} d_Z^2(z_1, z_2) \, \mathrm{d}\pi(z_1, z_2) \Big)^{\frac{1}{2}} \\
&= \Big( \int_{Z_1 \times Z_2} d_Z^2(z_1, z_2) \, \mathrm{d}\pi(z_1, z_2) + \lambda \sum_{i=1}^{2} \underbrace{\mathrm{GW}(\mathbb{X}_i, (Z, d_Z, P_{i,\sharp}\pi))}_{=0} \Big)^{\frac{1}{2}} \\
&\leq \lim_{n \to \infty} \Big( \int_{Z_1 \times Z_2} d_Z^2(z_1, z_2) \, \mathrm{d}\pi_n(z_1, z_2) + \lambda \sum_{i=1}^{2} \mathrm{GW}(\mathbb{X}_i, (Z, d_Z, P_{i,\sharp}\pi_n)) \Big)^{\frac{1}{2}} \\
&\leq \lim_{n \to \infty} \mathrm{EW}_{\lambda_n}(\mathbb{X}_1, \mathbb{X}_2) \leq \mathrm{EW}(\mathbb{X}_1, \mathbb{X}_2).
\end{aligned}
$$

Hence $\pi \in \Pi(I_{1,\sharp}\xi_1, I_{2,\sharp}\xi_2)$ and

$$
\mathrm{EW}(\mathbb{X}_1, \mathbb{X}_2) = \inf_{J_i : X_i \hookrightarrow Z, i=1,2} \mathrm{W}(J_{1,\sharp}\xi_1, J_{2,\sharp}\xi_2) = \Big( \int_{Z_1 \times Z_2} d_Z^2(z_1, z_2) \, \mathrm{d}\pi(z_1, z_2) \Big)^{\frac{1}{2}}
$$

implies that $(I_1, I_2)$ minimizes $\mathrm{EW}(\mathbb{X}_1, \mathbb{X}_2)$ and $\pi$ realizes $\mathrm{W}(I_{1,\sharp}\xi_1, I_{2,\sharp}\xi_2)$. $\qquad \square$

**Proof of Proposition 3.5**  Without loss of generality, assume that $\pi_n$ converges to $\pi \in \Pi(\zeta_1, \zeta_2)$ with $\zeta_i \in \mathcal{P}(Z)$ weakly; otherwise take a convergent subsequence. Since $(Z, d_Z)$ is compact, the diameter $M := \sup_{z,z' \in Z} d(z, z')$ is finite. Using plans $\tilde{\pi}_n \in \Pi(\zeta_{1,n}, \zeta_{2,n})$, where $\zeta_{i,n} \in \mathcal{P}(Z)$ are arbitrary GW approximations of $\mathbb{X}_i$, i.e., minimizers of (4), we estimate the relaxed embedded Wasserstein metric as follows:

$$
\mathrm{GW}^2(\mathbb{X}_1, (Z, d_Z, P_{1,\sharp}\pi_n)) + \mathrm{GW}^2(\mathbb{X}_1, (Z, d_Z, P_{2,\sharp}\pi_n)) \leq \frac{1}{\lambda_n} \mathrm{EW}_{\lambda_n}^2(\mathbb{X}_1, \mathbb{X}_2)
$$

$$
\leq \frac{1}{\lambda_n} \int_{Z \times Z} d_Z^2(z, z') \, \mathrm{d}\tilde{\pi}_n(z, z') + \sum_{i=1}^{2} \mathrm{GW}^2(\mathbb{X}_i, (Z, d_Z, P_{i,\sharp}\tilde{\pi}_n)) \leq \frac{M^2}{\lambda_n} + \mathrm{GWA}(\mathbb{X}_1) + \mathrm{GWA}(\mathbb{X}_2).
$$

Taking the limit $n \to \infty$, and exploiting (Beier & Beinert, 2025, Lem. I.1), we notice that the left-hand side becomes $\mathrm{GW}^2(\mathbb{X}_1, (Z, d_Z, \zeta_1)) + \mathrm{GW}^2(\mathbb{X}_2, (Z, d_Z, \zeta_2))$. Hence $\zeta_i$, $i = 1, 2$, have to be GW approximations of $\mathbb{X}_i$ on $(Z, d_Z)$. $\quad \square$

**Proof of Proposition 3.6**  Without loss of generality, assume that $\pi_n$ converges to $\pi \in \mathcal{P}(Z \times Z)$ weakly; otherwise take a convergent subsequence. Estimating $\mathrm{EW}_{\lambda_n}$ using the plans $\tilde{\pi}_n := (\mathrm{Id}, \mathrm{Id})_{\sharp}\zeta_n$, where $\zeta_n \in \mathcal{P}(Z)$ is an arbitrary fixed-support barycenter, i.e., a minimizer of (5), we obtain

$$
\mathrm{EW}_{\lambda_n}^2(\mathbb{X}_1, \mathbb{X}_2) \leq \lambda_n \sum_{i=1}^{2} \mathrm{GW}^2(\mathbb{X}_i, (Z, d_Z, P_{i,\sharp}\tilde{\pi}_n) = \lambda_n \, \mathrm{GWB}(\zeta).
$$

Thus, $\lambda_n \to 0$ implies $\mathrm{EW}_{\lambda_n}(\mathbb{X}_1, \mathbb{X}_2) \to 0$ and therefore $\mathrm{W}^2(P_{1,\sharp}\pi_n, P_{2,\sharp}\pi_n) \to 0$. Due to the stability of the Wasserstein distance (Ambrosio et al., 2005, Prop. 7.1.3.), the limit of $\pi_n$ has the form $\pi = (\mathrm{Id}, \mathrm{Id})_{\sharp}\zeta$ for some $\zeta \in \mathcal{P}(Z)$. Moreover, we have

$$
\mathrm{GW}^2(\mathbb{X}_1, (Z, d_Z, P_{1,\sharp}\pi_n)) + \mathrm{GW}^2(\mathbb{X}_1, (Z, d_Z, P_{2,\sharp}\pi_n)) \leq \frac{1}{\lambda_n} \mathrm{EW}_{\lambda_n}^2(\mathbb{X}_1, \mathbb{X}_2) \leq \mathrm{GWB}(\mathbb{X}_1, \mathbb{X}_2).
$$

Taking the limit $n \to \infty$, and exploiting (Beier & Beinert, 2025, Lem. I.1), we notice that the left-hand side becomes $\mathrm{GW}^2(\mathbb{X}_1, (Z, d_Z, \zeta)) + \mathrm{GW}^2(\mathbb{X}_1, (Z, d_Z, \zeta))$. Hence $\zeta$ has to be a fixed-support GW barycenter. $\quad \square$

**Proof of Proposition 3.7** For arbitrary $\alpha \in \mathcal{P}(X_1 \times Z_1 \times Z_2 \times X_2)$, and for $\pi := P_{Z_1 \times Z_2, \sharp}\alpha$ and $\gamma_i := P_{X_i \times Z_i, \sharp}\alpha$, the functional in (9) can be estimated by

$$
F_\lambda(\alpha) = \iint_{(X_1 \times Z_1 \times Z_2 \times X_2)^2} \left[ \tfrac{1}{2}\big(d_Z^2(z_1, z_2) + d_Z^2(z_1', z_2')\big) + \lambda \sum_{i=1}^2 (d_{X_i}(x_i, x_i') - d_Z(z_i, z_i'))^2 \right] \mathrm{d}\alpha(x_1, z_1, x_2, z_2)\, \mathrm{d}\alpha(x_1', z_1', z_2', x_2')
$$

$$
= \int_{Z_1 \times Z_2} d_Z^2(z_1, z_2)\, \mathrm{d}\pi(z_1, z_2) + \lambda \sum_{i=1}^2 \iint_{(X_i \times Z_i)^2} (d_{X_i}(x_i, x_i') - d_Z(z_i, z_i'))^2\, \mathrm{d}\gamma_i(x_i, z_i)\, \mathrm{d}\gamma_i(x_i', z_i')
$$

$$
= \int_{Z_1 \times Z_2} d_Z^2(z_1, z_2)\, \mathrm{d}\pi(z_1, z_2) + \lambda\,(G_{X_1,Z}(\gamma_1) + G_{X_2,Z}(\gamma_2)) \geq \mathrm{EW}_\lambda^2(\mathbb{X}_1, \mathbb{X}_2). \tag{15}
$$

Now let $\pi^*$ and $\gamma_i^*$ be solutions of (6). Due to the gluing lemma (Villani, 2003, Lem. 7.6), there exists $\alpha^* \in \mathcal{P}(X_1 \times Z_1 \times Z_2 \times X_2)$ such that $P_{Z_1 \times Z_2, \sharp}\alpha^* = \pi^*$ and $P_{X_i \times Z_i, \sharp}\alpha^* = \gamma_i^*$. For this $\alpha^*$, the inequality in (15) becomes sharp, i.e., $F_\lambda(\alpha^*) = \mathrm{EW}_\lambda(\mathbb{X}_1, \mathbb{X}_2)$. Thus $\alpha^*$ solves (9). The other way round, let $\alpha^*$ be a solution of (9). As argued in the first part of the proof, the inequality in (15) has to be again sharp. Hence $\pi^* := P_{Z_1 \times Z_2, \sharp}\alpha^*$ and $\gamma_i^* := P_{X_i \times Z_i, \sharp}\alpha^*$ solve (6). $\square$

## B. Numerical Studies on GW, EW, and EW$_\lambda$

Our approach enables geometrically meaningful data comparison as we can approximate the EW metric on general metric spaces $Z$. We validate this in two experiments.

**Approximation of** EW **by** EW$_\lambda$ **for Synthetic Circular Data** Consider circular mm-spaces $\mathbb{X}_i = (\mathbb{S}^1, d_{\mathbb{S}^1}, \xi_i)$ with $(Z, d_Z) = (\mathbb{S}^1, d_{\mathbb{S}^1})$, where $(\mathbb{S}^1, d_{\mathbb{S}^1})$ denotes the circle equipped with a circular distance. We set $\xi_1$ uniformly distributed and $\xi_2$ according to the density of a von Mises distribution with increasing dispersion parameter $\kappa$. Note that the von Mises distribution takes the form of a uniform distribution for $\kappa = 0$ and becomes more concentrated for $\kappa \to \infty$. Due to the rotational invariance of the uniform distribution on the circle, the embedded Wasserstein distance can easily be calculated in this case as it coincides with the Wasserstein distance. Thus, we can ignore the isometry in EW and directly calculate EW by solving the linear program underlying the OT problem. We discretize the circle into 360 bins and estimate EW$_\lambda$ for different choices of $\kappa$ and $\lambda$. The results in Figure 8 show an improved approximation of EW for larger $\lambda$ as announced in Proposition 3.4. Indeed, we have an excellent fit for $\lambda = 20$, whereas $\lambda = 0.2$ leads to an underestimation.

**2d Shape Matching** Next, we compare randomly rotated gray-value images. We can describe such images as mm-spaces $\mathbb{X}_i = (X, d, \xi_i)$ where $X \subset \mathbb{R}^2$ is a grid that describes the pixel positions equipped with the Euclidean metric and $\xi_i$ is the pixel intensity, see (Beier et al., 2023). We use $(Z, d_Z) = (X, d)$. We apply random affine transformations, i.e., translations and rotations, to the first 10 FashionMNIST (Xiao et al., 2017) training images from the classes "Trouser", "Pullover" and "Sneaker", respectively. Then we compute W, GW and EW$_\lambda$ for $\lambda = 20$. The results are displayed in Figure 9. As expected, the Wasserstein distance cannot recover the class structure due to the affine transformations, whereas EW$_\lambda$ and GW capture the class structure. Moreover, we see that the distance matrices of EW$_\lambda$ and GW display almost identical patterns, up to scaling.

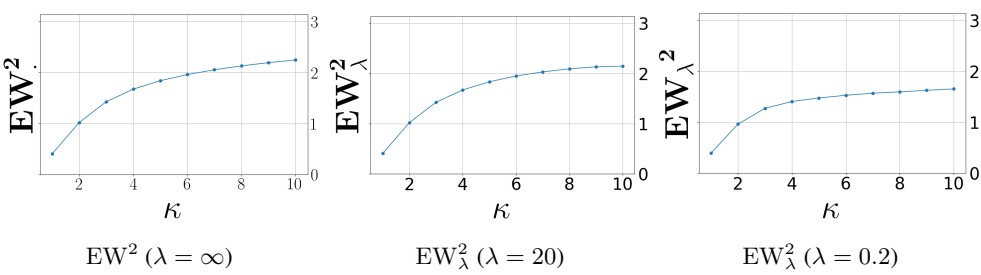

$\mathrm{EW}^2\ (\lambda = \infty)$ \qquad $\mathrm{EW}_\lambda^2\ (\lambda = 20)$ \qquad $\mathrm{EW}_\lambda^2\ (\lambda = 0.2)$

*Figure 8.* Comparison of EW with EW$_\lambda$ for different $\lambda$. While the curves for EW (left) and EW$_{20}$ (middle) are almost identical, those for EW$_{0.2}$ (right) is consistently smaller.

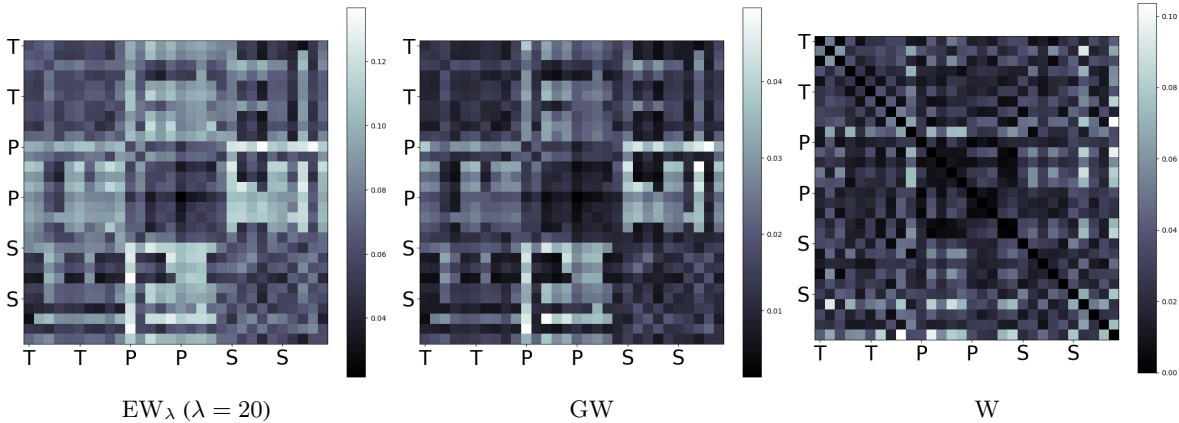

EW$_\lambda$ ($\lambda = 20$)       GW       W

*Figure 9.* Pairwise W, GW and EW$_{20}$ distances of 10 randomly rotated and translated images from each of the "Trouser" (T), "Pullover" (P), and "Sneaker" (S) FashionMNIST classes. Smaller distances are darker. The class block structure is recovered with EW$_\lambda$ and GW, but not with Wasserstein.

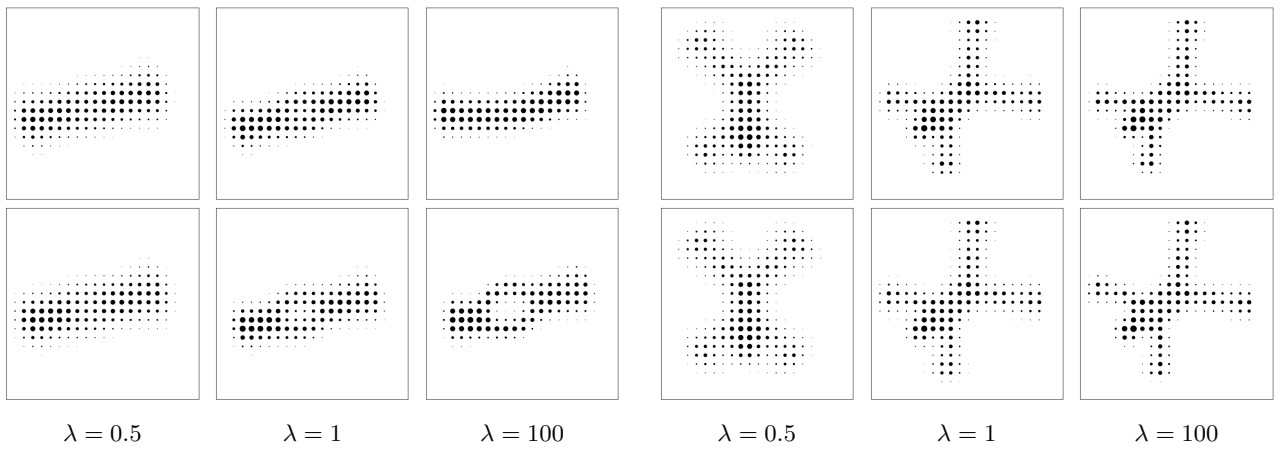

$\lambda = 0.5$    $\lambda = 1$    $\lambda = 100$    $\lambda = 0.5$    $\lambda = 1$    $\lambda = 100$

(a) S-bended rectangle and Swiss roll with hole.    (b) Same human shape in different poses.

*Figure 10.* Ablation study of $\lambda$ for joint embedding of shapes from Figure 4b and 5a. Based on varying $\lambda$ values, we employ our method with $\varepsilon = 0.001$ and $Z$ defined by a uniform $20 \times 20$ grid in $[0, 1.3]^2$.

## C. Parameter Sensitivity

Algorithm 1 depends on three input parameters, namely the GW regularization parameter $\lambda$, the entropic regularization parameter $\varepsilon$ and the chosen discrete reference space $Z$. Generally, we aim to set $\lambda$ as high as possible and $\varepsilon$ as low as possible while preserving numerical stability. Based on two sets of 3d shapes described in Section 6.1, namely a s-curve in combination with a Swiss roll with a hole and two human shapes, we present a parameter sensitivity study that illustrates the impact of $\lambda$, $\varepsilon$ and $Z$. Throughout this study note that our Euclidean joint embeddings are invariant to translations and rotations which results in arbitrarily oriented joint embeddings. In Figure 10, we investigate the influence of the GW parameter $\lambda$ that promotes near-isometric embeddings. We observe that the embeddings of both spaces nearly coincide for small $\lambda$. This is supported by Proposition 3.6 showing that the embeddings form a fixed-support barycenter in the limiting case. Moreover, we see a saturation effect where an increase of $\lambda$ from 1 to 100 has only a small impact on the joint embeddings. In Figure 11, we investigate the influence of the entropic parameter $\varepsilon$ that enables efficient computation. Here, we see that a large $\varepsilon$ leads to a highly blurred embedding. This shows that it is advisable to choose a low, but numerically stable value for $\varepsilon$. Lastly, we investigate the discretization of the reference space in Figure 12. Again, we aim for a fine discretization, but the runtime of our algorithm depends heavily on the size $|Z|$ of our reference space. We observe that the joint embeddings display the same shapes and alignments, only at higher resolution. All embeddings were computed with 40 iterations of Algorithm 1.

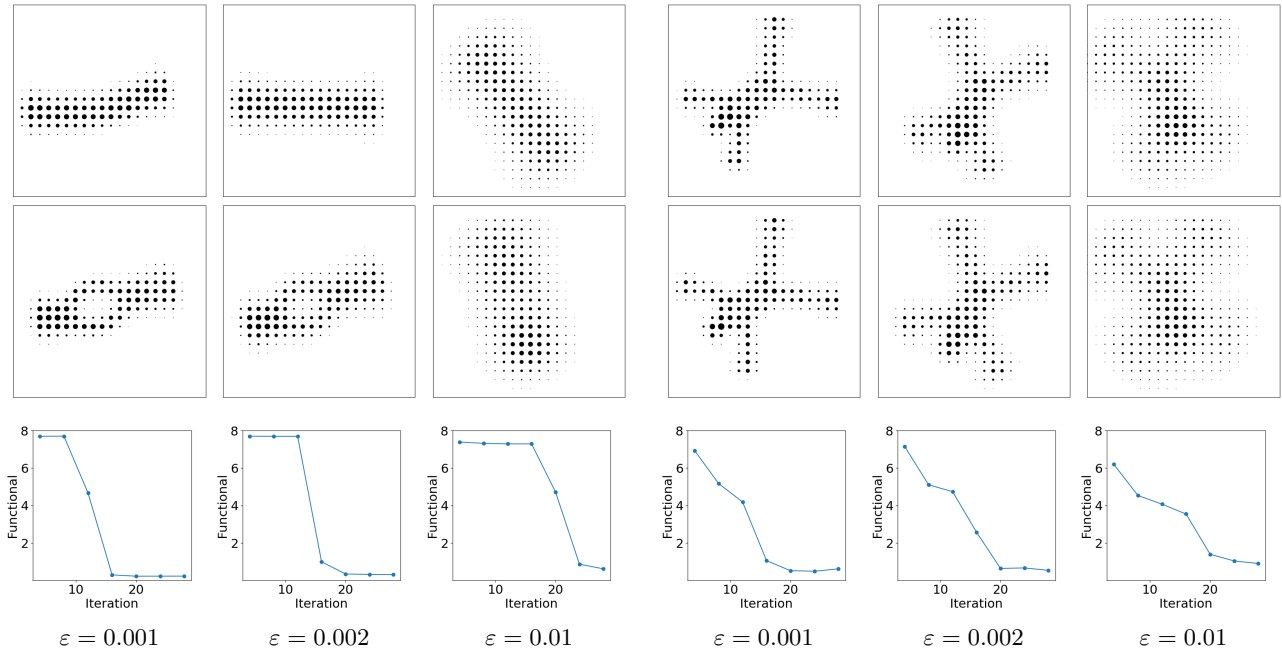

| $\varepsilon = 0.001$ | $\varepsilon = 0.002$ | $\varepsilon = 0.01$ | $\varepsilon = 0.001$ | $\varepsilon = 0.002$ | $\varepsilon = 0.01$ |

(a) S-bended rectangle and Swiss roll with hole.      (b) Same human shape in different poses.

*Figure 11.* Ablation study of $\varepsilon$ for joint embedding of shapes from Figure 4b and 5a. Based on varying $\varepsilon$ values, we employ our method with $\lambda = 100$ and $Z$ defined by a uniform $20 \times 20$ grid in $[0, 1.3]^2$. Below the joint embeddings, we show convergence plots of the unregularized functional $F_{100}(\alpha)$ from (9) after 4, 8, 12, 16, 20, 24 and 28 iterations.

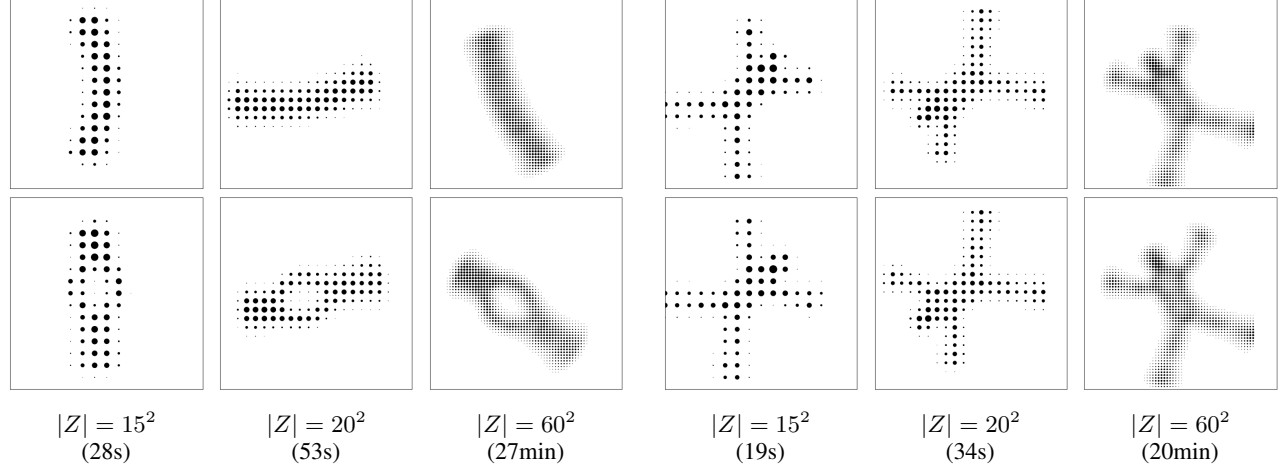

| $|Z| = 15^2$ (28s) | $|Z| = 20^2$ (53s) | $|Z| = 60^2$ (27min) | $|Z| = 15^2$ (19s) | $|Z| = 20^2$ (34s) | $|Z| = 60^2$ (20min) |

(a) S-bended rectangle and Swiss roll with hole.      (b) Same human shape in different poses.

*Figure 12.* Ablation study of the grid size for joint embedding of shapes from Figure 4b and 5a. Based on $\varepsilon = 0.001$ and $\lambda = 100$, we employ our method with $Z \subset [0, 1.3]^2$ defined by a uniform $15 \times 15$, $20 \times 20$ or $60 \times 60$ grid. We include CPU runtimes.

## D. Hyperparameter Selection

JMDS, SCOT, and our method all rely on the Sinkhorn algorithm. Therefore, we use the same fixed entropic regularization $\varepsilon$ for all three methods for a given experiment. We choose it as small as possible while avoiding numerical overflow, i.e., $\varepsilon = 0.001$ or $\varepsilon = 0.002$. Next, there are multiple ways of defining the nearest neighbor graph described in Section 6.2. For our experiments, we start by running the SCOT algorithm which offers an unsupervised hyperparameter search to determine the shape of the nearest neighbor graph (Demetci et al., 2022). We use the unsupervised hyperparameter selection of the

publicly available SCOT implementation[3] to estimate a suitable nearest-neighbor graph. Here, we use the same $\varepsilon$ as in the rest of the experiment and use the default grid over the neighborhood size. Note that the public implementation uses a connectivity graph based on pairwise correlation distances between points instead of Euclidean distances, see (Demetci et al., 2022). Then, we use SCOT and MDS to estimate the final embeddings based on the pairwise geodesic distance matrix estimated from this graph. We use the same distance graph for UnionCom, JMDS, and our algorithm. Here, we estimate suitable hyperparameters based on a random 10% validation split. We use four parameter configurations for each algorithm, i.e., we test $\lambda = 0.01, 0.1, 10, 100$ for JMDS and our method. For UnionCOM, we use combinations of $\beta = 1, 10$ and a perplexity of 30 and 100 for the t-SNE dimensionality reduction. We use no parameter annealing for any method.

## E. Further Example for Comparing Feature Spaces

As an addition to the genotyping experiment, we present another experiment with real-world data based on latent spaces. We consider the task of comparing the 4d latent spaces of an auto-encoder (AE) versus a variational auto-encoder (VAE), trained on FashionMNIST (Xiao et al., 2017). For this purpose, we train a simple convolutional AE with two convolutional layers and one linear layer for the encoder and the decoder. We train the AE with the MSE loss and the VAE with the log-likelihood loss for 10 epochs on the canonical training data splits. Subsequently, we embed the test data using the resulting AE and VAE into the 4d latent spaces. Note that such distinct AE trained on the same dataset generally produce incomparable embeddings. For our experiment, we consider the first 100 data points of the test split. We choose $Z$ as an equispaced $20 \times 20$ grid on $[0, 1.3] \subset \mathbb{R}^2$ with the Euclidean distance. We follow the same evaluation routine as in Subsection 6.2 for parameter selection and visualization. The joint embeddings are visualized in Figure 13. Comparing FOSCTTM and KNN, we see again a good embedding quality for our approach, see the figure caption.

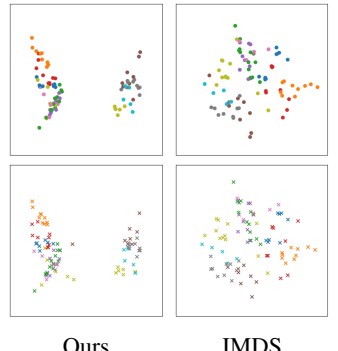

Ours        JMDS

*Figure 13.* Joint embedding of the FashionMNIST latent space of an AE and a VAE into $\mathbb{R}^2$ using our method and JMDS. The AE latent space is on the top and the VAE latent space is on the bottom. Both methods align the color-coded classes. A quantitative comparison shows that our model achieves a better FOSCTTM (0.040) than JMDS (0.042), whereas our KNN-Accuracy (0.589) is slightly worse than the one of JMDS (0.644).

---

[3]https://github.com/rsinghlab/SCOT (March 2025)

