# OpenReview forum: "Joint Metric Space Embedding by Unbalanced Optimal Transport with Gromov–Wasserstein Marginal Penalization"
_ICML.cc/2025/Conference — ICML 2025 poster_

### Official Review · Reviewer_CPxj · 2025-03-12

**Overall Recommendation:** 2

**Summary:**

In this paper the authors propose a novel alignment procedure  procedure across different spaces by optimizing a Wasserstein distance in a joint embedding space between two distributions that are close  in the GW sens to two distributions.  The idea is similar to the Sturm variant of GW and the authors express the problem as the optimization of the weighs of the distributions on a fixed support (calling it GW marginal penalization). They reformulate the problem as a multi-marginal OT problem and  propose a block coordinate descent algorithm to solve the optimization using at each step unbalanced MMOT solvers. A nice connection between the proposed approach and the existing Joint-MDS is discussed (joint MDS estimated support in euclidean space while the proposed approach estimate weights at fixed support). Several proof of concept of the approach illustrate what can be done with it in practice with notably very good performances in single cell alignment.

## update after rebuttal

Thanks to the authors for their rebuttal. It clarified a few questions and the new results are appreciated but some proposed changes in the paper cannot be properly reveiwed so I will keep my score.

**Claims And Evidence:**

The claims in the abstract are all reasonable and well supported. The empirical evidence about the interest of the method is a bit limited since as stated it relies a lot on several proof-of concept but they illustrate well the method and the diversity of potential application.

**Essential References Not Discussed:**

An important reference that should be discussed because the proposition of estimating the weighs minimizing GW on a fixed support is illustrated in their Figure 2: The proposed approach is a clear extension that handle multiple distributions instead of one but the connection is clear and more general metrics on Z are proposed in the reference

Van Assel, Hugues, et al. "Distributional Reduction: Unifying Dimensionality Reduction and Clustering with Gromov-Wasserstein." arXiv preprint arXiv:2402.02239 (2024).

The low rank OT methods use a similar final factorization between the OT plans (where the rank r is the size oif the support in Z for the proposed method). For instance Forrow can be seen as a Wasserstein version of the joint MDS.

Scetbon, Meyer, Gabriel Peyré, and Marco Cuturi. "Linear-time gromov wasserstein distances using low rank couplings and costs." International Conference on Machine Learning. PMLR, 2022.

Forrow, Aden, et al. "Statistical optimal transport via factored couplings." The 22nd International Conference on Artificial Intelligence and Statistics. PMLR, 2019.

Halmos, Peter, et al. "Low-rank optimal transport through factor relaxation with latent coupling." Advances in Neural Information Processing Systems 37 (2024): 114374-114433.

A very similar the MMOT formulation with  specific structure was proposed in :
Kerdoncuff, Tanguy, et al. "Optimal tensor transport." Proceedings of the AAAI Conference on Artificial Intelligence. Vol. 36. No. 7. 2022.

**Experimental Designs Or Analyses:**

The experiments are interesting. The visualizations are pertinent and show the interest of the method. On the other hand by doing multiple proofs of concept, the authors limit their study and barely discuss or illustrate the impact of some very important parameter such as the effect of lambda and of the choice of the fixed support Z_i.

Also the results on the single cell alignment are interesting but not really reproducible without the code since no parameter validation grids are provided.

**Methods And Evaluation Criteria:**

As stated above the numerical experiments are proof of concept and rely  mostly on qualitative visualizations expect for the application to single cell alignment where classical performance measures for the field are reported (and the proposed approach use a very specific approach using barycenter mapping.)

**Other Comments Or Suggestions:**

See above

**Other Strengths And Weaknesses:**

+ The method is interesting, the "projection" onto a joint space and the use of Wasserstein is an intriguing way to transfer information between different source distribution in different spaces and to recover a global alignment through the shared space.

+ The limit case of the proposed approach should be discussed more in detail. if lambda goes to infinity then the solution and the most isometric preserving projection that minimize the Wass distance (but the two projections P_#\pi can be very different). When lambda goes to 0 then the problem seems to revert to a GW barycenter with fixed support since the Wasserstein must go to 0 so you search for a unique distribution that is close to the two in the GW sens hence a barycenter. Those special case are important to discuss (an prove maybe results?) so that one can have a better idea of the positioning wrt the very active OT GW extension community.

+ How extendable is the proposed approach to multiple distributions? The method is already quite heavy with a very high dimensional MMOT and does not seem to scale. This is not a problem but it needs to be discussed.

+ The proposed reformulation an optimization algorithm are also interesting. But more details about why there is an equivalence between the MMOT formulation and the factorized OT plan is important (due to the structure of the global objective value but not as obvious as stated).

+ The choice of the geometry and support in Z is very sensitive. The experiments are carefully selected so that either the geometry corresponds to the distribution (unroll in 2D, torus, 2D euclidean known to work well for single cell). But outside of the very toy example of sub-part of torus, the geometry is barely used in practice or remains euclidean so the argument that the proposed method is more general because it can be used for non euclidean geometry is not well justified in practice. Especially since learning the geometry can be done implicitly with estimation a GW barycenter that provides pairwise "distances" on the learned manifold and also allows projection on a smaller dimension in euclidean.

+ Numerical experiments are mostly proof of concepts. They are interesting and clearly the proposed strategy of fixing the support can be of interest wrt JMDS. But on the other hand it is a bit difficult to estimate the impact and practical use of the proposed method outside of single cell alignment (and even in this case it is compared only to few methods). This limits a little the potential impact of the method.

+ As in many OT papers the authors do a complete modeling of the problem they want to solve and at the end just add some entropy regularization when numerical optimization becomes necessary. This behavior should not be promoted since entropic regularization has an important impact on the final OT plans and a study of the impact of this regularization in terms of mass spread/performance VS computational time is necessary to evaluate its impact. As noted above this kind of parameter sensitivity analysis (also wrt lambda and the support of Z) is important for the practitioner.

**Questions For Authors:**

See above

**Relation To Broader Scientific Literature:**

The paper is obviously strongly related to recent extension  of MDS that is very well discussed and positioned. But some references from the OT community are missing and would deserved to be discussed.

**Theoretical Claims:**

The theoretical claim seem to be correct from a quick pass over the proofs in appendix but I did not check them in detail. The transition from the original problem to a multi-marginal problem is a bit quick and the proof of proposition 3.5 that is important in practice is missing so the supplementary would be improved with those details (that are not as obvious as stated in the paper).

---

> ### Author Rebuttal · Authors · 2025-03-31
>
> Many thanks for your positive report and the constructive comments.
> We revised our manuscript accordingly,
> especially with respect to the limiting cases
> and additional references.
> Please find our detailed answers below.
>
> **Limiting Cases:**
> Thanks for pointing this out,
> which extends the limiting case ($\lambda\to\infty$)
> for existing isometric embeddings
> discussed in Prop 3.4.
> As you suggested,
> the limit ($\lambda\to0$) actually yields
> a fixed support barycenter,
> and the limit ($\lambda\to\infty$ without isometries)
> gives
> the best possible projection
> for both spaces separately.
> In the revised version,
> we extend Section 3
> by adding a new proposition.
> The rigorous proof
> relies on a subsequence argument
> and the weak continuity
> of the EW functional
> similar to the proof of Prop 3.4.
> The proof is added to the supplement.
>
> **Extendability:**
> From a theoretical point of view,
> the proposed variational formulation can be adapted
> to handle multiple input spaces.
> The main issue hereby is
> how to compare the different projections on $Z$.
> Intuitively,
> we may rely on pairwise Wasserstein distances;
> but this approach does not yield
> a 'tree-like' cost function
> for the multi-marginal reformulation.
> As a consequence,
> the employed multi-marginal Sinkhorn method
> (Beier et al, 2022, doi:10.1007/s10851-022-01126-7)
> cannot be applied.
> An alternative would be to utilize
> only the Wasserstein distance
> between the $k$th and $(k+1)$st projection.
> Since the employed multi-marginal Sinkhorn method
> linearly scales in the number of spaces,
> we expect that the resulting method
> remains numerically tractable
> for small numbers of spaces.
> The actual choice of the model
> and the numerical implementation
> of a multiple space variant,
> however,
> requires further studies.
> Thanks for pointing out this imported extension,
> which is discussed
> in the conclusion
> of the revised manuscript.
>
> **Multi-Marginal Reformulation:**
> In the final version,
> the rewriting of the original problem is extended
> to give a better intuition behind the equivalence.
> The detailed proof of Prop 3.5 is added
> to the supplement.
>
> **Choice of Geometry**
> You are right:
> the geometry of $Z$ has to be chosen
> with respect to the problem at hand.
> To study the sensitivity
> with respect to the discretization of $Z$,
> we repeated the experiments in Section 6.1
> (unrolling, 2d embedding)
> with different grid sizes.
> The results remain stable
> meaning that we obtain the reported embeddings
> in different resolutions
> ([$Z$-Experiment](https://imgur.com/a/rlnnkVr)).
> The chosen grid,
> however,
> has a major impact on the computation time.
> These experiments are added to the supplement.
> Please notice
> that,
> besides the alignments on the sphere and torus,
> also the alignment of the Gaussian mixture models
> rely on a non-Euclidean geometry
> since the alignment is calculated
> in the space of Gaussians
> equipped with the Wasserstein distance itself.
> We agree that
> these experiments are only proof-of-concepts
> that can only point in the direction of possible applications
> and that real-world applications require further study.
>
> **Parameter and Regularization:**
> To study the impact of $\lambda$ and the entropic regularization,
> we extend the experiments in section 6.1.
> As you pointed out,
> small $\lambda$ leads to fixed support barycenters
> and large $\lambda$ to GW-minimizing projections
> of the input spaces
> ([$\lambda$-Experiment](https://imgur.com/a/g1Thyrg)),
> which nicely relates to the theoretical limiting cases.
> To study the impact of the regularization,
> we again adapt the experiments in Section 6.1
> ([$\varepsilon$-Experiment](https://imgur.com/a/odRSmkl)).
> The results are added to the supplement,
> and we discuss the impact of entropic regularization.
>
> **Code and Reproducibility:**
> Thanks for pointing out
> that the parameter ranges
> for the grid searches
> are missing.
> We use the same distance graph and the same entropic parameter
> $\varepsilon=0.001$ for all algorithms and four hyperparameter configurations per method.
> We test $\lambda=0.01,0.1,10,100$ for JMDS and our method. For UnionCOM, we use combinations of $\beta=1,10$ and
> a perplexity of 30 and 100 for the t-SNE dimensionality reduction.
> SCOT only relies on $\varepsilon$ and the input distances and has no additional hyperparameters.
> In the revised version,
> the ranges are added.
> Also,
> our implementations will be publicly available via Github
> to reproduce or reuse our implementations.
> Due to the anonymity requirement and
> a short e-mail exchange on the official guidelines,
> we cannot link the code here,
> but the link will be provided in the final version.
>
> **References:**
> Thanks for pointing out the related references
> which we incorporate in the revised version.
> Regarding the GW dimension reduction,
> we also refer to (Clark, Needham, Weighill: arXiv:2405.15959v2),
> where the relation between GW and MDS is discussed.
>
> We would like to thank you again for your suggestions
> that significantly improved the manuscript
> and hope that you can accept the paper now.

---

### Official Review · Reviewer_haA5 · 2025-03-14

**Overall Recommendation:** 3

**Summary:**

The authors formulate an unbalanced OT problem where the marginals are penalized with GW distances and prove that their function have the minimizer. The authors show that if the regularization parameter goes to infinity, the solution their optimization problem is the embedded Wasserstein distance. To approximate the solution to their unbalanced OT problem, they define a relaxed bi-convex relaxation and present an algorithm using block-coordinate descent and multi-marginal entropy regularized OT. As their formulation seeks near-isometric joint embeddings into a metric space, the authors can use their method to find joint embeddings of data with different feature spaces. They utilize their method for some artificial 3D shape datasets and to align single cell sequencing data.

### Update after rebuttal

Thanks to the authors for their responses. I keep my score

**Claims And Evidence:**

The claims in the submission are supported.

**Essential References Not Discussed:**

I do not know of any essential references not discussed.

**Experimental Designs Or Analyses:**

I checked the experiments. I am not quite sure what to make of the results in section 6.1. It seems like the results are in summarized Figure 4 and Figure 5 and they are more qualitative. Is there a way to compare the performance of JMDS against the author's method on the FAUST dataset across a larger set of pairs of human poses? Are the results in Figure 5 true across all pairs of poses in FAUST? Additionally, while Table 1 shows some convincing performance gains for their method over JMDS, I am not sure what the message of Figure 6 is? I do not see a very clean separation of cell types in either JMDS or the author's method.

**Methods And Evaluation Criteria:**

The evaluation criteria make sense for the problem at hand.

**Other Comments Or Suggestions:**

1. The authors consistently use the notation $I_{1, \sharp} \xi_1$ which I believe is the push-forward measure but I think this notation is probably not familiar to everyone in the ICML audience so the authors might want to make a note.

2. Grammatical error on line 136: "the following relaxation enables to handle arbitrary...." There should be another word between "enables" and "to".

**Other Strengths And Weaknesses:**

Strengths: The framework that the authors present seems to be novel as they produce their metric embeddings by defining a unbalanced OT problem with Gromov-Wasserstein penalization, which has not been previously done. They also connect their work to previous work as they show their optimization problem is the same as a previously defined "embedding Wasserstein metric" as the regularization term goes to infinity.

Weaknesses: The experiments seem to be a bit thin, as I mentioned above.

**Questions For Authors:**

1. Do we know how quickly algorithm 1 converges?

2. Are there any guarantees about the quality of the approximation that Algorithm 1 produces?

**Relation To Broader Scientific Literature:**

This paper seems to be most closely related to the joint multi-dimensional scaling algorithm which also uses OT to produce embeddings of different datasets into a shared Euclidean space. The authors provide a new framework for joint metric embeddings by using \emph{unbalanced} OT and Gromov-Wasserstein distance (to enforce the embedding space to be nearly isometric to the inputs).

**Theoretical Claims:**

I did not check the proofs.

---

> ### Author Rebuttal · Authors · 2025-03-31
>
> Many thanks for the positive evaluation
> and your constructive feedback!
> We have revised our manuscript according to your suggestions in particular
> with respect to quantitative experiments.
> Please find our detailed answers to your questions below.
>
> **Quantitative FAUST:**
> We agree that
> the results shown in Figure 4 and 5
> are qualitative
> and can only give fleeting impressions
> of the algorithms.
> Since our method and JMDS can both be derived
> from the same functional,
> we compare the achieved objectives,
> the Wasserstein distances between embeddings,
> and the GW distances between embeddings and input spaces.
> The experiment is repeated
> for each pairwise combination
> of the first ten shapes from the FAUST dataset
> and different $\lambda$.
> For comparison,
> the shapes are downsampled to 35 vertices,
> and both methods are regularized with $\varepsilon=0.001$.
> For our method, we set $Z$ according to a uniform
> 30$\times$30 grid on $[0, 1.3]$.
> The average and standard deviation of
> the GW distance (input space to embedding)
> and Wasserstein distance (between embeddings)
> are as follows: [Quantitative FAUST](https://imgur.com/a/x3EVFwa).
> In total,
> we here achieve better alignments than JMDS both in terms of GW and Wasserstein.
> While our method requires 4-5 seconds per embedding, JMDS requires 2-3 seconds.
> These results
> are added to the revised version.
>
> **Separability:**
> We agree that
> the classes in Figure 6 are not clearly separated.
> Nevertheless,
> the constructed 2d embeddings increases the KNN accuracy
> compared with the other methods.
> The message behind Figure 6 is that
> the embeddings of the two datasets
> from completely different feature spaces
> looks quite similar
> and that the classes are mapped to common regions,
> i.e.,
> the embeddings have similar color codings.
> To improve the visual embeddings,
> we now equip the feature spaces
> with distances from
> the unsupervised SCOT distance estimation routine
> (Demetci, et al, doi:10.1089/cmb.2021.0446).
> The updated distances lead to
> the following [new version](https://imgur.com/a/U1E7GL7) of Figure 6.
>
>
> **Experiments:**
> Please note that there are additional experiments in the supplement
> (comparison between GW, $\textrm{EW}_\lambda$ and Wasserstein in Appendix B;
> domain adaption experiment in Appendix C).
> To further improve the numerical section,
> we add the above quantitative study of the FAUST dataset.
> Moreover, we add illustrative
> supplementary sensitivity studies
> with respect to
> the relaxation parameter ([$\lambda$-Experiment](https://imgur.com/a/g1Thyrg)),
> the entropic parameter ([$\varepsilon$-Experiment](https://imgur.com/a/odRSmkl))
> and the discretization of the joint space  ([$Z$-Experiment](https://imgur.com/a/rlnnkVr)).
>
>
> **Minor Comments:**
> - Please notice that the push-forward is introduced
>   in the first paragraph of Section 2.
>   To improve the presentation,
>   we set the definition in
>   its own equation.
> - The grammatical error is corrected now.
>
> **Questions:**
> - In our experiments,
>   Alg 1 needs less than 50 iterations to converge.
>   In the revised version,
>   we show the convergence of the objective over iterations
>   to our $\varepsilon$ sensitivity study
>   ([$\varepsilon$-Experiment](https://imgur.com/a/odRSmkl)).
> - Unfortunately,
>   there is so far no theoretical guarantee
>   about the convergence of Alg 1,
>   which we hope to find in our future work.
>
> We hope that we met all your concerns in the revised manuscript.

---

### Official Review · Reviewer_FyDV · 2025-03-18

**Overall Recommendation:** 3

**Summary:**

This paper proposes a framework for jointly embedding and aligning two metric measure spaces into a common metric space. The authors build their formulation from well-known prior works in this area, namely the Sturms and Memolis variants of Gromov Wasserstein distance. The main contribution of this paper is focused on a relaxation that can handle arbitrary metric measure spaces (i.e., not necessarily isomorphic) resulting in a relaxed Embedded Wasserstein Distance $EW_\lambda$. This ultimately results in an unbalanced optimal transport problem with Gromov–Wasserstein marginal penalization. The authors provide a derivation of a numerical approach to solving it using bi-convex relaxation.

The experiments are mostly synthetic and focus mainly on comparing with a conceptual counterpart - JMDS Chen et.al 2023. Specifically, the results are shown for joint embeddings of (1.) 3D shapes - FAUST, Swiss Roll, spherical and toroidal embeddings, (2.) Feature spaces and synthetic GMMs.

The key difference seems to be that JMDS applies a free support approach for obtaining the coordinates in the joint embedding whereas the proposed $EW_\lambda$ opts for a fixed support instead and optimizing for the weights. Multiple proof of concept demonstrations validate this aspect and show aligned, meaningful joint embeddings of the metric spaces

**Claims And Evidence:**

Yes. I think the paper is theoretically constructive in its structure and new definitions are usually followed by a derivation of essential properties.

**Essential References Not Discussed:**

No

**Experimental Designs Or Analyses:**

Yes.

**Methods And Evaluation Criteria:**

I think the experiments are comprehensive and a good proof of concept with multiple figures showing good joint embeddings. However, I do miss a broader applicability of the approach with respect to say applications like non-rigid 3D shape matching, transfer learning, domain adaptation, etc.

**Other Comments Or Suggestions:**

Overall, I found this to be an interesting submission. It is theoretically comprehensive and conceptually enriches many prior OT-based distances. However, I won't categorize it with a clear acceptance - since it could be revised structurally for much more impact. Furthermore,  the experiments are much too synthetic and without any "outstanding" demonstrations clearly showing the real benefit of $EW_\lambda$ over others. Also, I am missing any computational complexity issues like runtime, memory, etc.

**Other Strengths And Weaknesses:**

This paper does feel lacking in a couple of aspects

- The writing and structure are not optimal and somewhat unpolished. e.g. I am not sure I understand and agree with the comment on Page 1 "However, classic OT lacks invariance to important transformations like rotations in Euclidean space",
- Line 297, typoJMDS and not JMSD,
- I found Figure 3 very difficult to parse. What does it convey and why is the specific coloring?

- Little discussion on the complexity of the method in comparison to other GW-based distances

**Questions For Authors:**

Figures 4 and 5 why are the embeddings of the proposed method not centered unlike JMDS?

**Relation To Broader Scientific Literature:**

To the best of my knowledge - I think the paper has been positioned quite well. The remarks on Lines 295 - 307 on Page 6 were the most refreshing: clearly outlining the key differences with JMDS. All things considered, I will not fault this submission too much for lack of novelty.

**Theoretical Claims:**

The important ones like the definition and properties of $EW_\lambda$

---

> ### Author Rebuttal · Authors · 2025-03-31
>
> Many thanks for your positive evaluation of our manuscript as an 'interesting submission' that 'is theoretically comprehensive and conceptually enriches many prior OT-based distances'
> and for your constructive feedback!
> We revised our manuscript according to your suggestions.
> Please find our answers below.
>
> **Experiments:**
> We agree that
> the shown experiments are mainly proof-of-concepts.
> To improve the numerical part,
> we extend Section 6.2 by a [CPU runtime comparison](https://imgur.com/a/RwHMZXm)
> with respect to the different methods.
> Moreover,
> we extend the supplement by a
> numerical sensitivity analysis
> with respect to the relaxation ([$\lambda$-Experiment](https://imgur.com/a/g1Thyrg))
> and regularization parameters
> ([$\varepsilon$-Experiment](https://imgur.com/a/odRSmkl))
> as well as
> with respect to the discretization of the joint space
> ([$Z$-Experiment](https://imgur.com/a/rlnnkVr)).
> Please also note the domain adaptation experiment in Appendix C.
> Thanks for pointing out possible future applications
> which, at the moment, are beyond the scope of this paper.
>
>
> **Complexity:**
> In comparison to GW,
> the complexity of our method is increased
> since we rely on a multi-marginal OT problem.
> However,
> please note that
> GW gives only a correspondence between the measures,
> but no direct information about a joint embedding.
> As mentioned above,
> we extend the feature space embedding experiment
> to compare the [runtime](https://imgur.com/a/RwHMZXm)
>  between the proposed approach
> and other methods in the literature.
> Moreover,
> we now study the runtime in dependence of the discretization of $Z$
> for the 2d embeddings in Section 6.1 ([$Z$-Experiment](https://imgur.com/a/rlnnkVr)).
>
> **Structure:**
> - Comment on page 1:
>   With the lack of invariance
>   (with respect to shapes),
>   we mean that the computed Wasserstein plans
>   depend on the actual embeddings of the shapes.
>   In contrast,
>   the Gromov-Wasserstein plans are always
>   independent of the embeddings or representations
>   of the considered shapes.
>   In the revised version the comment is accordingly clarified.
> - Figure 3:
>   The figure illustrates the multi-marginal plan $\alpha$,
>   which is, in fact, four-dimensional.
>   The labeled marginals $\gamma_1$, $\gamma_2$, and $\pi$
>   give the relation to the optimization problem
>   behind the original EW formulation.
>   The colors match the coloring of Fig 2,
>   which means that
>   everything related to $\mathbb{X}_1$ is red and
>   to $\mathbb{X}_2$ blue.
>   We add this in the captures of the figures.
> - Lastly, we will look into improving the structure and writing for the final manuscript.
>
> **Questions:**
> The reason why our embeddings are not centered is that
> we show the push-forward to the complete chosen grid $Z$.
> For JMDS,
> we obtain two point clouds
> and manually select a region containing both,
> which leads to a 'centered' visualization.
> In fact,
> our and JMDS embeddings are invariant to (joint) translations.
>
> We hope that we have addressed your concerns.

---

### Official Review · Reviewer_cuNW · 2025-03-23

**Overall Recommendation:** 4

**Summary:**

The paper proposes an unbalanced optimal transport problem with Gromov-Wasserstein marginal penalization, which can map data from two different domains, where no known correspondences exist, into a common metric space. The problem can be resolved using block-coordinate descent based on the Sinkhorn algorithm. A series of experiments, including joint embedding of 3d shapes, alignment of feature spaces, and alignment of Gaussian mixture models, have verified the effectiveness of the proposed EW_lambda distance.

## Update After Rebuttal
Thank you for the authors' response, which has resolved my confusion. I keep my score.

**Claims And Evidence:**

The proofs of Propositions 3.1-3.5 related to the formulation and solution of EW_lambda are clear, and the block-coordinate descent method based on Sinkhorn is reasonable. However, I think there is a lack of comparison with the GW distance, including aspects such as computational complexity, runtime, convergence, and the alignment effect.

**Essential References Not Discussed:**

I think the relevant literature has been covered.

**Experimental Designs Or Analyses:**

The experiment is reasonable, and the effectiveness of EW_lambda can be seen in various alignment experiments. Is it necessary to have a comparison with the baselines based on GW distance?

**Methods And Evaluation Criteria:**

The experiments on joint embedding of 3D shapes, alignment of feature spaces, and alignment of Gaussian mixture models are reasonable and can verify the capability of joint embeddings in both Euclidean and non-Euclidean spaces.  However, I'm curious as to why there was no comparison made with the alignment based on the GW distance.

**Other Comments Or Suggestions:**

There are no other suggestions here.

**Other Strengths And Weaknesses:**

Pros:

The paper presents the EW_lambda method which can map data from two distinct domains without known correspondences into a common metric space. It also theoretically proves the existence of a minimizer for this problem. Moreover, a bi-convex relaxation based on Sinkhorn is employed to numerically solve the issue. Additionally, the relationship with JMDS is discussed and the effectiveness of the method is experimentally validated.

Cons:

(1) There is a lack of in-depth discussion on how to select the lambda parameter according to different datasets/features. The sensitivity analysis of the regularization parameter should be analyzed.

(2) There are uncertainties about whether there are limitations when this method is extended to large datasets or high-dimensional features.

(3) There is a comparison with GW in the method section, but there may be a lack of GW-based alignment baselines in "Numerical Results" section.

(4) The bi-convex relaxation and block coordinate descent methods are based on the Sinkhorn algorithm. Are there any cases of numerical instability for this algorithm in practical applications? If so, how can these issues be addressed?

(5) From the results of Figure 6, compared to JMDS, the proposed method fails to effectively represent the clustering structure of the dataset.

**Questions For Authors:**

All the questions are in "Other Strengths And Weaknesses".

**Relation To Broader Scientific Literature:**

Embedding different domains into a common space is beneficial for comparing different features, and it has practical application value. The  proposed method seeks near-isometric joint embeddings of the inputs into a metric space, which relaxes the constraints compared with Sturm's GW distance.

**Theoretical Claims:**

I have reviewed the proofs of Propositions 3.1-3.5 and implementation of Algorithm 1, and I think they are reasonable.

---

> ### Author Rebuttal · Authors · 2025-03-31
>
> Many thanks for your positive and constructive feedback!
> We have revised our manuscript according to your suggestions.
> Please find our detailed answers to your questions below.
>
> **GW Comparison:**
> If we consider the experiment behind Fig 1,
> then we are looking for rotations (isometries)
> which aligns the shapes/measures on the sphere/torus.
> A classical GW plan gives only a correspondence
> between the measure supports,
> but no direct information
> how to rotate the whole sphere/torus
> such that the supports are aligned.
> To say it briefly, GW cannot directly solve the considered task.
> Although not directly related to the embedding problem,
> GW and EW are compared in Fig 9 (Appendix B)
> for 2d shapes from FashionMNIST.
> Here the values of both distances strongly correlate.
>
> **Parameter Selection:**
> To study the impact of $\lambda$,
> we extend the experiments
> in Section 6.1.
> The results have been added to the supplement
> ([$\lambda$-Experiment](https://imgur.com/a/g1Thyrg)).
> Figuratively,
> small $\lambda$ leads to fixed support barycenters,
> whereas large $\lambda$ gives the best possible embeddings
> even when no isometries exist.
> The two limiting cases $\lambda \to 0$ and $\lambda \to \infty$
> are also supported theoretically.
> The proofs,
> which rely on similar techniques than Prop 3.4,
> are added to the supplement.
>
> **High-Dimensional Features**
> The dimension of the feature space
> only affects the computation of the input distance matrices
> and not our algorithm directly.
> However, the size of the dataset
> and the chosen discretization of the joint space $Z$
> directly affects the runtime ([$Z$-Experiment](https://imgur.com/a/rlnnkVr)).
>
> **Bi-Convex Relaxation:**
> Unfortunately,
> we have no theoretical guarantee
> that the proposed algorithm converges.
> We choose a small, but numerically stable $\varepsilon$.
> In practice,
> we usually observe convergence after less than 50 iterations.
> However,
> in rare cases,
> the algorithm can get stuck
> in local minima.
> We add this discussion and a convergence experiment
> in the revised supplement
> ([$\varepsilon$-Experiment](https://imgur.com/a/odRSmkl)).
>
> **Clustering:**
> Please notice that
> Fig 6 serves mainly as an illustration
> of how the datasets are embedded
> and depends on the employed input distance.
> Looking at the color coding,
> we see that
>  the classes are located in the same areas.
> Although the actual visual separation is not as clear as for JMDS,
> the classification accuracy is increased.
> To improve the visual embeddings,
> we now equip the feature spaces
> with distances from
> the unsupervised SCOT distance estimation routine
> (Demetci, et al, doi:10.1089/cmb.2021.0446).
> The updated distances lead to
> the following [new version](https://imgur.com/a/U1E7GL7) of Figure 6.
>
> We hope that we have addressed your questions.

---

### Decision · Program_Chairs · 2025-05-01

**Decision:**

Accept (poster)

**Comment:**

In this submission, the authors proposed a new OT-based method to embed and align incomparable geometric objects jointly, leading to an unbalanced OT (UOT) problem with a GW-based marginal penalty. The proposed method leads to an optimization problem solved by block coordinate descent. Experimental results show the potential of the proposed method for aligning Euclidean and non-Euclidean objects.

The scores of the proposed method are mixed, but most reviewers appreciate the idea and admit the novelty of the method. However, AC and some reviewers have concerns about the computational efficiency of the method, especially when dealing with large-scale and high-dimensional objects. These concerns are not fully resolved in the rebuttal phase.

In summary, AC recommends to "weakly accept" this work.